# Revisiting the soil carbon saturation concept to inform a risk index in European agricultural soils

T. S. Breure [1] ✉, D. De Rosa [2], P. Panagos [1], M. F. Cotrufo [3], A. Jones[1] & E. Lugato [1]

The form in which soil organic carbon (SOC) is stored determines its capacity and stability, commonly described by separating bulk SOC into its particulate- (POC) and mineral-associated (MAOC) constituents. MAOC is more persistent, but the association with mineral surfaces imposes a maximum MAOC capacity for a given fine fraction content. Here, we leverage SOC fraction data and spectroscopy to investigate POC/MAOC distribution, together with SOC changes data over 2009–2018 period, across pedo-climatic zones in the European Union and the UK. We find that rather than a universal mineralogy-dependent maximum MAOC capacity, an emergent effective MAOC capacity can be identified across pedo-climatic zones. These findings led us to propose the SOC risk index, combining SOC changes and effective MAOC capacity. We find that between 43 and 83 Mha of agricultural soils are classified as high risk, mostly constrained to cool and humid regions. The index provides a synthetic information to decision makers for preserving and accruing POC and MAOC.

The pathway to climate neutrality foresees the contribution of the land to offset the residual sectorial greenhouse gas emissions by 2050, incrementing the carbon (C) removal from vegetation and soil. In the European Union (EU), operative policy instruments to increase the land C sink are atmospheric carbon dioxide ($CO_2$) removal targets in the land use, land use change and forestry (LULUCF) regulation[1] as well as the recent Carbon Removal Certification regulation[2], including carbon farming. Agricultural soils in the EU, in particular, are depleted in soil organic carbon (SOC) as compared to other land uses[3]. Furthermore, the majority of EU agricultural soils are far from saturation of the stable mineral-associated organic carbon (MAOC) fraction[4,5], allowing the storage of additional C by changing to appropriate management practices[6,7]. However, a recent data-driven study estimated a relative SOC loss of 0.75% for the period 2009–2018 in European agricultural soils[8]. These SOC losses occurred despite the introduction of both mandatory and voluntary schemes in 2013, aiming at increasing agricultural sustainability[9].

Assessing current bulk SOC content and its change over time (ΔSOC), while fundamental, does not provide enough information for effective SOC sequestration interventions. In the last decades, a new conceptual framework has highlighted the advantage to separate bulk SOC in two fractions that underlie prevailing mechanisms of SOC formation and stabilization, namely the MAOC and the particulate organic carbon (POC)[10]. MAOC is mostly composed of plant and microbial derived compounds low in molecular weight, which can be stabilized by interaction with the soil matrix via sorption and physical protection[11]. Consequently, MAOC is more resilient to degradation compared to POC, and it has a lower turnover time on average[4] which promotes the long-term accrual of atmospheric $CO_2$ into soil. However, MAOC has a 'theoretical mineral capacity' due to a finite number of mineral surface binding sites, as postulated and demonstrated by a large body of studies[12–14]. Therefore, the degree of MAOC saturation indicates the proportion of measured MAOC over the theoretical capacity. The theoretical mineral capacity is commonly calculated

[1]European Commission, Joint Research Centre, Ispra, Italy. [2]Department of Agriculture, Forestry, Food and Environmental Sciences, University of Basilicata, Potenza, Italy. [3]Department of Soil and Crop Science and Natural Resource Ecology Laboratory, Colorado State University, Fort Collins, USA. ✉e-mail: timo.breure@ec.europa.eu

based on soil texture and clay mineralogy to benchmark the saturation deficit of soils from global databases[15,16]. Here, we argue that this mineralogical capacity has a low practical importance for carbon accrual actions as, for instance, Mediterranean soils would never reach the MAOC content of acidic soils under the cold climate of northern Europe, even when sharing the same texture[17,18]. Commonly, the method used to calculate a unifying theoretical mineral MAOC capacity consists of pooling data together from different soil types, environmental and management conditions. Then, a linear regression is applied between MAOC and the soil's fine fraction content separately for high and low activity minerals (e.g. Georgiou et al.[16]). However, this approach does not acknowledge that the theoretical mineral capacity may not be achievable as MAOC storage is constrained by additional emerging ecosystem properties[19] that regulate SOC formation and stabilization such as pH, microbiome characteristics, type of litter and plant productivity[4,20–24]. The theoretical mineral capacity has also been questioned by recent studies[25], suggesting oversaturation of mineral particles due to the binding of organic matter to other organic matter bonded to minerals, therewith posing the fundamental question to what degree of surface loading MAOC can be still considered as "stabilized" by mineral-association[26].

Based on these premises, we use a clustered approach to calculate the 'effective MAOC capacity'. We followed Stewart et al. (2007), in that an apparent saturation limit can be reached since pedo-climatic and management conditions impose constraints even with increased C inputs[18]. The effective MAOC capacity in our clustered approach thus constitutes the biophysically achievable MAOC given the cluster's pedo-climatic properties, for soils under agricultural land use (Supplementary Fig. 1). Further, to account for the oversaturation of mineral particles[25,27], we formulated three different regression methods to estimate the effective MAOC capacity.

Additionally, we leveraged information from four datasets to map a risk index (Supplementary Fig. 1): i) the SOC content in locations that have been repeatedly surveyed (2009-2018) in the EU Land Use and Land Cover Survey (LUCAS)[28,29], ii) the SOC changes (ΔSOC) between the repeated surveys[8], iii) associated visible- and near-infrared (VNIR) spectroscopy measurements[30], and iv) a subset of measured SOC fractions[3,27]. The risk index builds on the exposure-vulnerability-hazard concept from the Intergovernmental Panel on Climate Change[31],

(Fig. 1). The exposure component consists of the areal extent, which are all soils under agricultural land use. The hazard is represented by ΔSOC, which is the effect of climate and management on SOC storage. Vulnerability is represented by the level of MAOC saturation within biophysically homogeneous European agricultural regions. We suggest that mapping the vulnerability and hazard components of agricultural SOC is informative for SOC management. While we applied this conceptual framework to the EU, which may be further refined with additional data, we suggest to apply it in other regions to identify areas at risk of SOC loss as well as areas with the highest potential for SOC accrual.

## Results and discussion

### Clustering of pedo-climatic zones across Europe

Bulk SOC storage is known to be an ecosystem property controlled by climatic conditions, management, plant productivity, soil properties such as texture and pH, and geomorphological features such as elevation or slope[19,21,23]. Therefore, approaches using pedo-climatic clustering can provide reliable estimates of bulk SOC storage, as recently demonstrated across Europe[32].

Similarly to bulk SOC, fractions vary with environmental, geochemical and landform gradients[22,24]. Thus, accounting for these conditions by applying a clustering approach can enable more accurate estimation of the effective MAOC capacity[24]. We therefore applied a k-means clustering procedure based on aridity[33], net primary productivity (NPP)[34], measured pH in $H_2O$[28] and landform[35] for the LUCAS soil sampling locations. Soil pH was included as a proxy of clay mineralogy and SOC turnover (microbial composition), landform to account for how the erosion and depositional setting affects preferential displacement of SOC fractions, NPP as a driver of saturation through C inputs[36] and aridity as a synthetic climate parameter controlling SOC storage[37–39]. We identified sixteen pedo-climatic clusters such as the coastal areas in mid- and southern-Europe (cluster 1) (Fig. 2), which generally receive high precipitation rates and show a large net primary productivity (Fig. 2b).

Relatively arid Mediterranean areas were attributed to separate clusters (3, 4 and 13), depending on their differences in landform, whereas their pH range was comparable. Other characteristic pedo-climatic zones were temperate lowland areas and the acid soils in north-western Europe (clusters 2, 5 and 15; Fig. 2). Pedo-climatic clusters also varied across smaller geographical scales. For example, considering the island of Sardinia (IT), the coastal areas were separated from the inland which showed further variation depending on the landform and soil pH (Fig. 2a).

### MAOC and POC predictions and total SOC changes in agricultural soils

Based on a subset of measured C fractions[3], we predicted POC and MAOC for the remainder of the LUCAS 2009 survey using visible near-infrared (VNIR) soil spectra. The VNIR spectra allowed for an independent estimate of POC and MAOC from the covariates used in the pedo-climatic clustering. Predicted MAOC showed good correspondence with measured values for the validation dataset, although to a lesser extent for POC (Supplementary Fig. 2). However, predicted POC and MAOC (i.e. POC + MAOC) showed relatively good correspondence with measured bulk SOC (Supplementary Fig. 3a), considering the number of samples and geographic extent of the LUCAS survey ($R^2 = 0.59$, RMSE = 8.8 g kg$^{-1}$, RPIQ = 1.7, Bias = 0.48, CCC = 0.76). We related the MAOC:SOC ratio to predicted carbon changes (ΔSOC) between the 2009–2018 surveys based on De Rosa et al. When plotting ΔSOC versus the MAOC:SOC ratio, pedo-climatic clusters showed different ranges for both variables (Fig. 3). Based on a linear least-squares model, all clusters showed a positive slope, where the interaction term of ΔSOC x Cluster was significant for different slope estimates (Supplementary Fig. 4, Supplementary Table 1–2). The positive

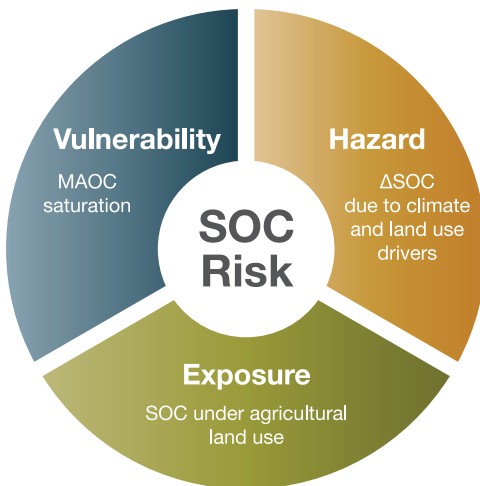

**Fig. 1 | Soil organic carbon (SOC) risk framework.** based on the exposure-hazard-vulnerability risk concept from the Intergovernmental Panel on Climate Change (IPCC)[31]. Soil organic carbon under agricultural land use is considered exposed. Vulnerability, the mineral-associated organic carbon (MAOC) saturation, determines the magnitude of the exposure. The hazard (soil organic carbon changes, ΔSOC) is the integrated effect of climate change and land use that acts on exposed SOC.

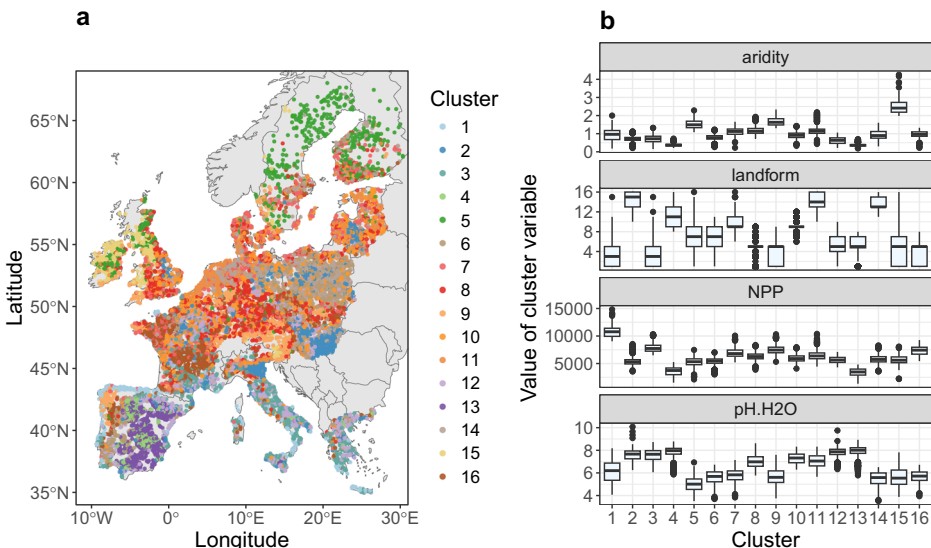

**Fig. 2 | Allocated clusters based on the *k*-means method using the Hartigan and Wong (1979) algorithm. a** Geographical representation of the LUCAS 2009 survey. **b** Boxplots of aridity, landform, net primary productivity (NPP) and soil pH used for the *k*-means in their original scale, by cluster association (*n* = 13,295). Vector map data used from the 'rnaturalearth' R package[81]. Copyright (CC0) (2025), (CRAN).

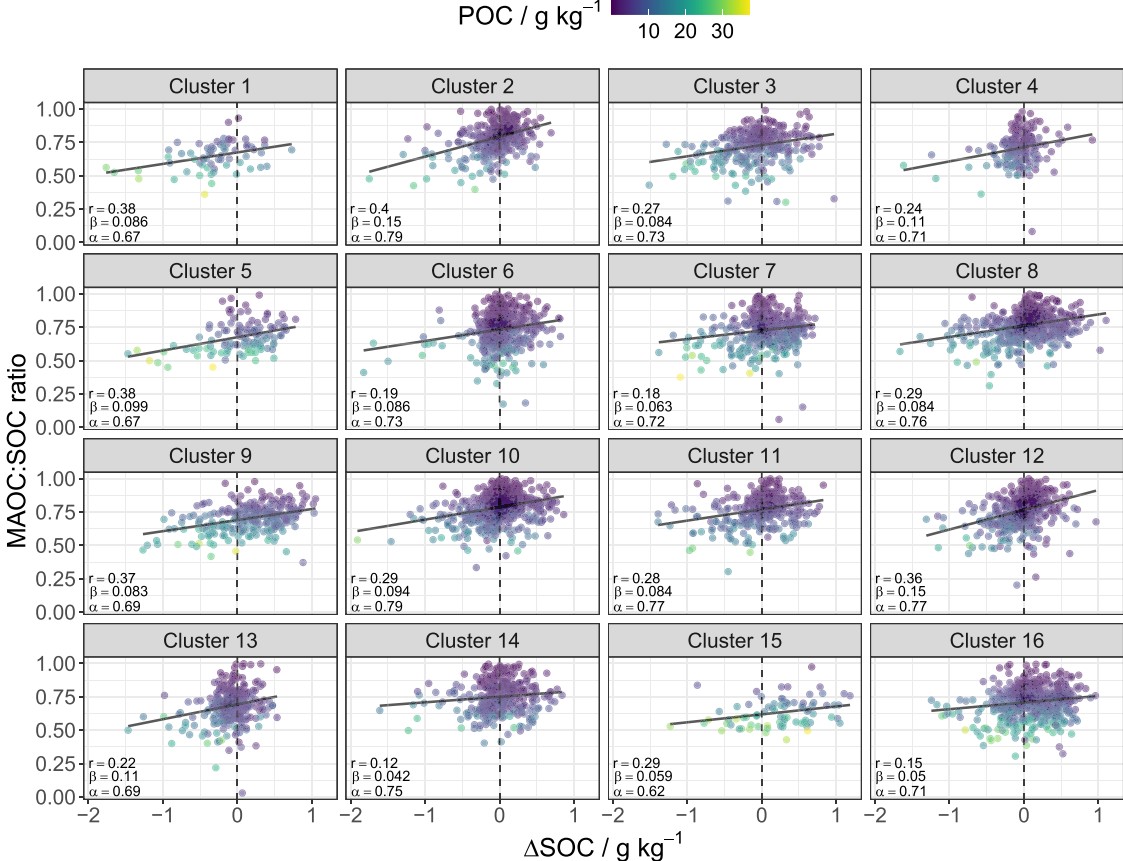

**Fig. 3 | Predicted changes in soil organic carbon (ΔSOC) versus the predicted mineral-associated organic carbon fraction (MAOC) of total soil organic carbon [MAOC:SOC ratio (with SOC = MAOC + POC)].** colored by the soil particulate organic carbon (POC) concentration for each pedo-climatic cluster. Cluster numbers are as in Fig. 2. *r* is the Pearson's correlation coefficient, α and β are the intercept and slope estimates of the linear least-squares model (solid black line). (*n* = 5482).

slope suggests that SOC losses (negative ΔSOC) were generally associated with both a higher contribution of POC to total SOC (i.e., low MAOC:SOC ratio) and a high POC content (g kg⁻¹ of soil; Fig. 3). These results are in line with previous findings, suggesting that POC is more vulnerable to disturbance than MAOC[3,38], and that MAOC is responsible for the largest amount of SOC increase from high quality inputs in agriculture[40,41]. Furthermore, cold and wet regions (clusters 5 and 15) had a lower MAOC:SOC ratio and high POC content (Fig. 3[39],). Data

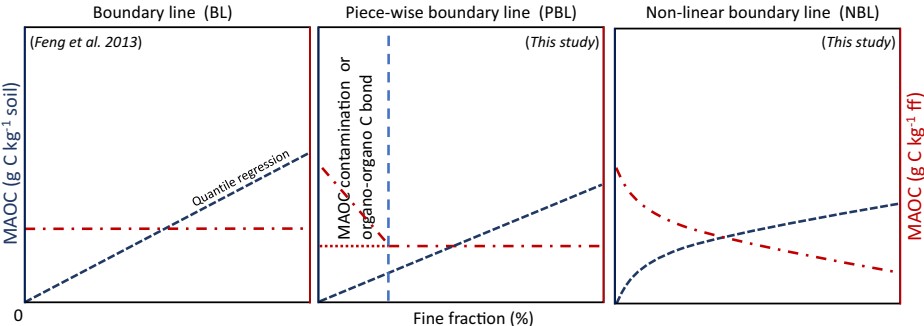

**Fig. 4 | Conceptual representation of different regression methods to estimate the effective mineral-associated organic carbon (MAOC) capacity.** Boundary line (**BL**) assumes that MAOC content (g C kg$^{-1}$ in fine fraction (ff)) has a constant maximum across the fine fraction range[15]. Piece-wise boundary line (**PBL**) assumes a constant for the effective MAOC capacity (like Feng et al.) but isolates the disturbance of MAOC contamination and organo-organo C bonds. The non-linear boundary line (**NBL**) assumes that MAOC content can change across the fine fraction range; effective MAOC capacity includes SOC organic layers that interact via organo-organo C bonds rather than directly to the clay surface. All regressions were done for the 90th quantile with a forced intercept to 0 as per Feng et al. Data and fitting statistics for the LUCAS survey is presented in Supplementary Figs. 5–7.

points with SOC gains (positive ΔSOC) generally showed higher MAOC:SOC ratios (Fig. 3; Supplementary Fig. 4), although SOC losses occurred also in soils with high MAOC:SOC ratio, due to varying type of perturbations driving SOC changes at the LUCAS sites.

## Estimating the effective MAOC capacity by clusters

Given that MAOC is less associated with SOC losses by perturbation than POC, efficient C management should better consider the degree to which soils are from their effective MAOC capacity, which we hypothesized being an emergent property of pedo-climatic clusters. We further formulated three different methods to estimate the effective MAOC capacity based on underlying MAOC saturation theory (Fig. 4).

The concept of MAOC saturation was first proposed by Hassink (1997) that described MAOC saturation as a linear function of the soil's fine fraction content (clay + silt content)[12]. Successively, Feng et al. proposed the boundary line (BL) regression as a method to better restrict the inference to data from soils close to MAOC saturation[15] (Fig. 4). Here we propose a variation of the BL method, which we called PBL, to better isolate soils that have mineral association between organic carbon and the fine fraction. Conversely, suggesting oversaturation of mineral particles due to the binding of organic matter to other organic matter bonded to minerals[25], we used a non-linear regression (NBL) to calculate the MAOC saturation (Fig. 4).

Following the different approaches illustrated, we performed quantile regressions to determine the cluster effective MAOC capacity (Supplementary Figs. 5–7). Parameter estimates varied by up to 200% between pedo-climatic clusters (Supplementary Figs. 5–7), which provides evidence that the effective MAOC capacity is an emergent property based on pedo-climatic conditions[19,24]. From the pedo-climatic variables used in the clustering, aridity and net primary productivity showed larger effects on the spread of parameter estimates compared to pH and landform. This suggests that aridity and NPP play a larger role in the distinction between the theoretical and effective MAOC capacity for our dataset (Supplementary Fig. 8), although the importance of pedo-climatic controls might vary across different regional scales[22].

However, the functional relationship of the BL method did not seem to fit the data well for most clusters, since the 90th quantile regression line under-fits for coarse soils (i.e., low in fine fraction content) and over-fits for soils high in fine fraction content (Supplementary Fig. 5). This fitting phenomenon has been reported previously in a study for German soils[5] and our study provides further evidence on a continental scale. The estimated MAOC capacity for the PBL method was less variable between clusters compared to the BL method. The estimated breakpoints, however, varied widely in their magnitude between clusters (19–68 %), depending on the level at which MAOC in

fine fraction reached a plateau (Supplementary Fig. 6). The non-linear NBL method allowed for a good fit of the upper boundary accounting for a slight increase in MAOC along the fine fraction range (Supplementary Fig. 7). Recently, Viscarra-Rossel et al. estimated the effective MAOC capacity across the Australian continent by soil groups. Their regression method is analogous to the underlying theory of our NBL method, in that it assumes a fine fraction-dependent MAOC concentration. To align with the existing literature, we have included estimates based on their method (Supplementary Fig. 9). These estimates can be considered an upper-limit of the effective MAOC capacity for our dataset, given that the frontier line method is specifically aimed at estimating the maxima of the data[24]. We preferred to use a more conservative quantile approach (see Method section), the parametric nature of which allows for comparison with previous studies.

Whereas two of our proposed regression methods to estimate MAOC saturation are novel to this study (Fig. 4), there are few studies that used the exact same regression type (90th quantile, 0 intercept) to determine MAOC saturation[42]. Here we compared our results to the existing literature. We first converted all parameter estimates to the same unit (g MAOC kg$^{-1}$ fine fraction). The mean of the β parameter estimates across clusters for the boundary line method (BL) was 45.1 ± 11.3 (SD) (Supplementary Fig. 5). The mean α value for the piecewise regression method (PBL) was 34.1 ± 6.6 (SD). Differences between the PBL method compared to the BL method occur because the MAOC concentration for soils low in fine fraction is assumed to be contaminated by POC or characterized by organo-organo C bonds[25,26] and, thus, identified before the estimated breaking-point in the piecewise regression (Supplementary Fig. 6). These results show that disregarding coarse soils with high MAOC content leads to lower estimates of the effective MAOC capacity. The mean estimate for the NBL method was 28.5 ± 6 (SD) and spanned the smallest range across pedo-climatic zones (Supplementary Figs. 5–7). The upper limit parameter values for the NBL and PBL methods (41, 45, respectively) were lower than for BL (62), which is lower than previous estimates for 2:1 mineral dominated soils: 84 ± 4 (SE) ([15], 90th quantile regression) and 86 ± 9 (90% CI) ([16], 95th quantile regression).

Given the LUCAS sampling design, our analysis is likely to be representative of the most abundant soil types across Europe[28]. However, the dataset that we used to calibrate VNIR spectra against C fractions to predict the 6,548 samples is relatively small and therefore can impose a limitation. For example, recent studies[42] pointed to soils with higher MAOC content that may be formed under particular conditions (e.g., very high clay, hydromorphic conditions) and can exceed 50 g MAOC kg$^{-1}$ soil; the maximum MAOC content in the C fraction dataset used in this study. For example, MAOC accumulation

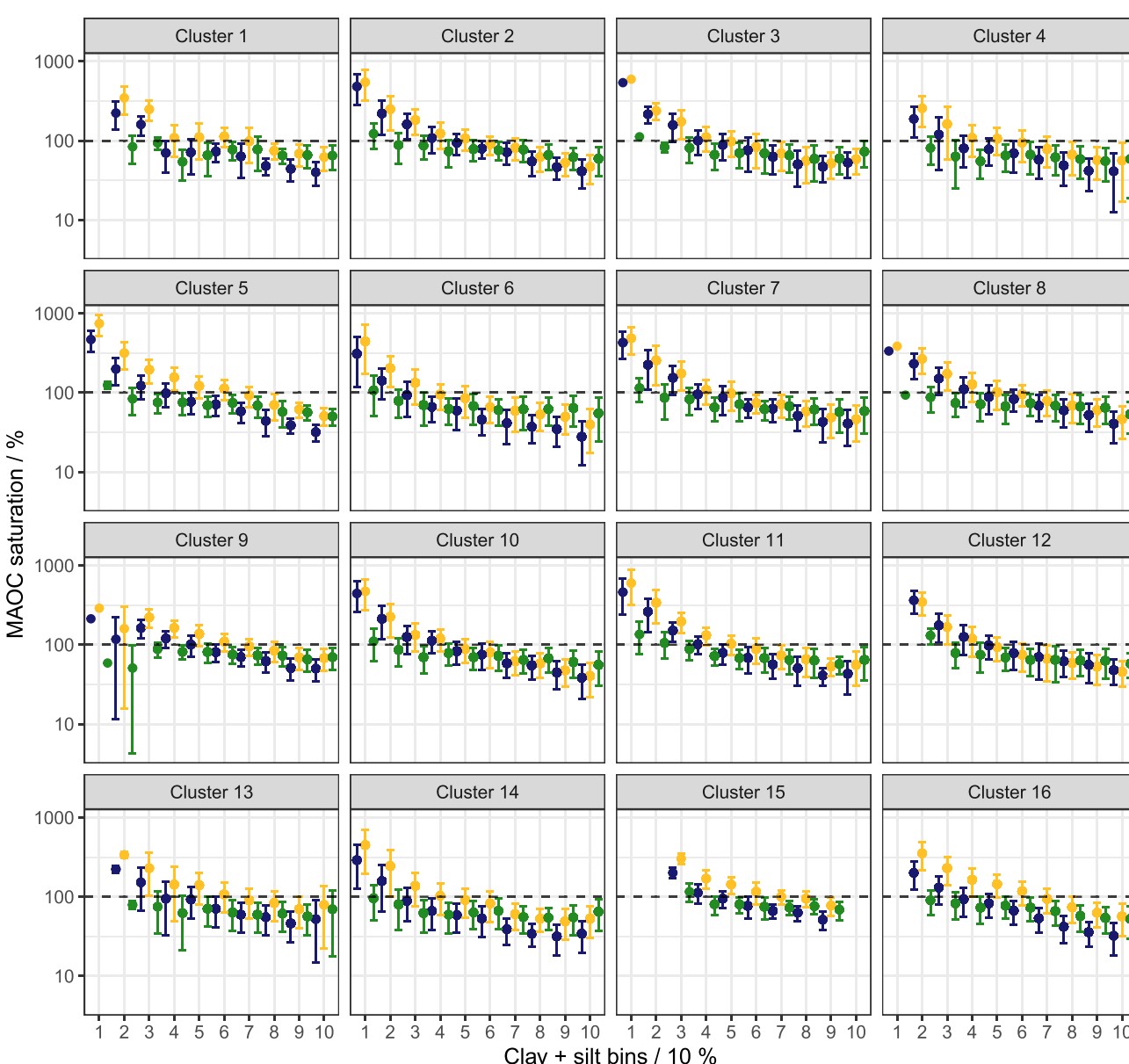

**Fig. 5 | Degree of mineral-associated organic carbon (MAOC) saturation (MAOC / effective MAOC capacity x 100%) as a function of fine fraction (clay + silt, %).** Values below 100% indicate a saturation deficit relative to the cluster-dependent effective MAOC capacity. Fine fraction content has been binned by intervals of 10%, points and lines represent the mean degree of MAOC saturation ± standard deviation for each bin. The y-axis is on a $\log_{10}$ scale. ($n$ = 6548).

can occur due to oxygen limitation rather than mineral stabilization[42], such as in Stagnosols. Another component that might interfere with MAOC accounting is the presence of geogenic C, i.e., the organic C present in the bedrock that was deposited during sedimentation[42,43]. These findings further support that a clustered approach is more meaningful for the inference of effective MAOC capacity from data spanning a broad range of soil types. That is, disaggregation prevents a limited number of points, potentially belonging to one particular soil type, from having high leverage in the regression. Future research could investigate how oxygen limited conditions and geogenic C affect regional estimates of MAOC saturation. Nevertheless, we have also investigated the effect of including the existing legacy soil C fraction data[5,16] on the parameter estimates (Supplementary Figs. 10–13). Based

on these results, we anticipate that the exclusion of soil under-represented in the LUCAS dataset might underestimate the maximum MAOC capacity for fine-textured soils in some pedo-climatic clusters. Nonetheless, the mean estimate across our EU pedo-climatic clusters was more similar to those found for 2:1 mineral dominated soils (global coverage) under cropland: 45 ± 5(SE)[15] (Supplementary Figs. 5–7). Also the estimates for non-clustered data align very closely with that of Feng et al. for 2:1 mineral dominated soils under cropland[15] (Supplementary Fig. 14). The NBL method notably led to lower estimates (29 g MAOC kg-1 fine fraction).

Similar differences between regression methods were found after calculating the degree (as percentage) of MAOC saturation (MAOC / effective MAOC capacity x 100%) and computing the mean and its

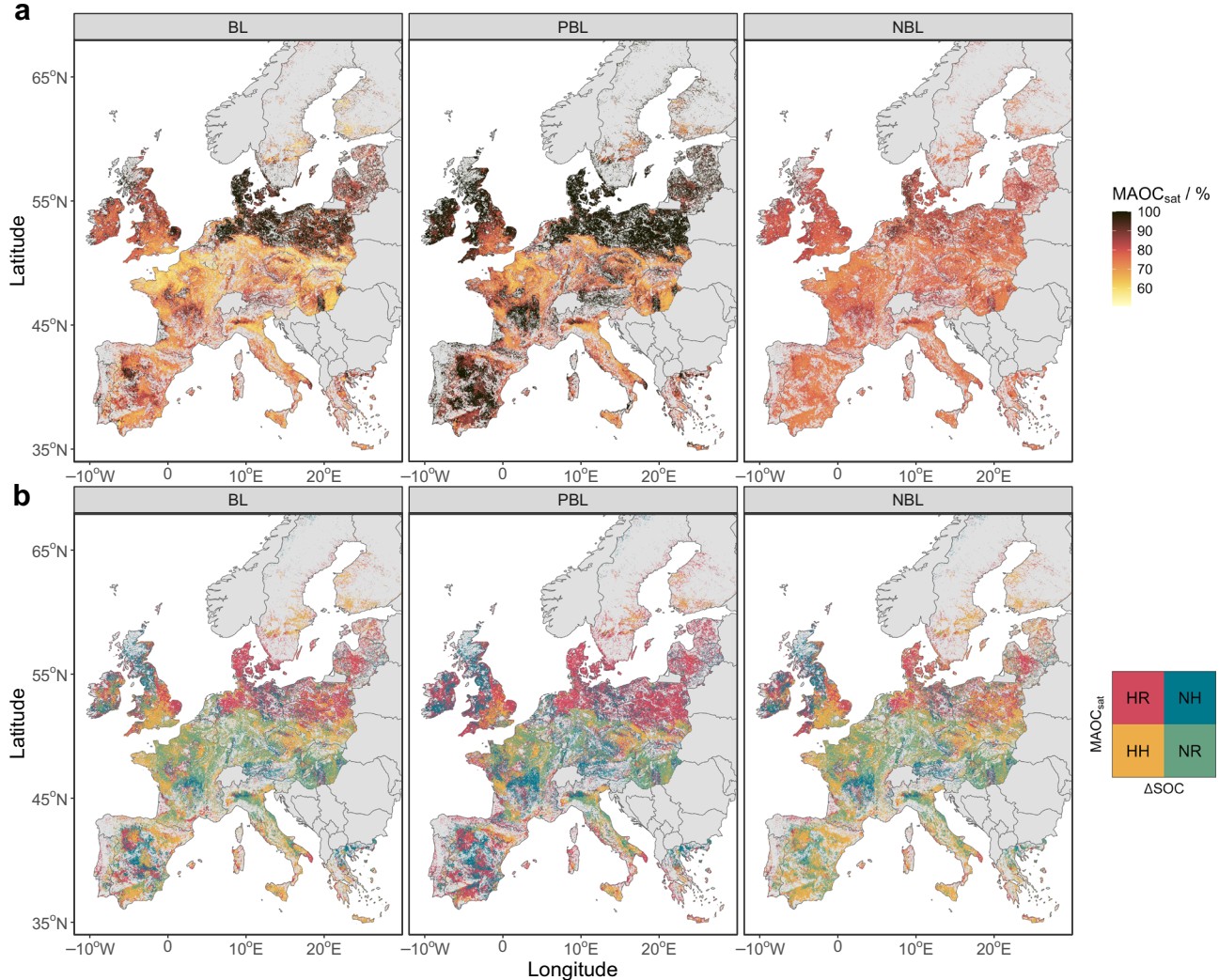

**Fig. 6 | Geographical representation of mineral-associated organic carbon (MAOC) degree of saturation (MAOC$_{sat}$) and the soil organic carbon (SOC) risk index. a** degree of MAOC saturation mapped to grid cells by cluster-specific MAOC saturation relationship for fine fraction bins. Values below 100% indicate a saturation deficit relative to the cluster-dependent effective MAOC capacity. Panels correspond to different methods to estimate the effective MAOC capacity (see Fig. 4). **b** follows the same panel structure and shows the SOC risk index:

whether MAOC saturation is above or below its median and whether ΔSOC is below 0 or not. 'HR': high risk, negative ΔSOC, MAOC saturation above median. 'HH': high hazard, negative ΔSOC and MAOC saturation below the median. 'NR': no risk, positive ΔSOC and MAOC saturation below the median. 'NH': no hazard, positive ΔSOC and MAOC saturation above the median. Vector map data used from the 'rnaturalearth' R package[81]. Copyright (CC0) (2025), (CRAN).

standard deviation for fine fraction intervals by cluster (Fig. 5). Values below 100% indicate a saturation deficit relative to the cluster-dependent effective MAOC capacity. The figures where the degree of MAOC saturation has not been binned by fine fraction intervals can be found in Supplementary Fig. 15.

When the BL or PBL was applied, MAOC content exceeded the theoretical saturation (>100%) in sandy soils likely because the quantile regression is underfitting the data low in fine fraction content. In total, 1202 samples exceeded 100% for BL and 1408 samples for PBL. For fine soils, the BL generally estimated lower MAOC saturation compared to the other methods. The NBL method did not frequently exceed 100% saturation (708 samples) and remained more constant across the range of fine fraction content compared to the other two methods. These characteristics can be attributed to a better fit across the range of fine fraction content, reducing variability in MAOC saturation at the extremes. However, we restricted MAOC saturation estimates to a maximum of 100% for subsequent analysis, under the assumption that any values above 100% indicate saturation.

## The SOC risk index

Since we demonstrated that effective MAOC saturation is cluster-dependent and interplaying with SOC vulnerability, here we propose a synthetic 'risk index', which may guide most effective action to protect or accrue SOC. We did this by borrowing the hazard-exposure-vulnerability risk framework used by the Intergovernmental Panel on Climate Change[31] (Fig. 1). Soil organic carbon under agricultural land use is considered exposed to anthropogenic and environmental drivers, and thus determines the areal extent of soils under exposure in the EU. The degree of MAOC saturation was taken as a measure of vulnerability, given that soils saturated in MAOC are more likely to either have a high SOC content and/or have proportionally higher POC. The SOC changes (ΔSOC) were considered as the level of hazard, that is, SOC changes driven by climatic conditions and land management[8]. By assessing the degree of hazard and vulnerability, we constituted four index classes (high risk, high hazard, no risk, no hazard) that allow for a spatial assessment of SOC status. High hazard (HH) and high risk (HR) are both subject to SOC losses but have low and high levels of vulnerability, respectively (MAOC saturation below or above the

median of 68.9%). No risk (NR) and no hazard (NH) have SOC gains with low and high vulnerability, respectively (Fig. 1, Fig. 6b).To assess the SOC risk index across Europe, we imposed the effective MAOC capacity as a function of the fine fraction for each pedo-climatic cluster (Method section, Supplementary Figs. 16–17). Upscaling the clusters thus allowed us to: i) map the degree of MAOC saturation across Europe and relate these estimates to ΔSOC and ii) reclassify both MAOC saturation and ΔSOC to assess the SOC risk index in agricultural soils.

The PBL method calculated a higher MAOC saturation in particular for the Baltic Sea area, northern UK and the Iberian Peninsula, although it followed the same geographical pattern as the BL method (Fig. 6a). This reflects that the PBL method estimates a lower effective MAOC capacity for coarse textured soils. That is, the abovementioned regions are characterized by low clay content and/or high sand. The NBL method was distinct compared to the two other approaches given its relative narrow range of MAOC saturation across Europe. For example, a smaller area is calculated to be MAOC saturated (mostly restricted to Denmark and north-east Germany) (Fig. 6a), likely due to a better fit for coarse soils leading them to have lower MAOC saturation (Fig. 5). The differences in MAOC saturation across Europe illustrate how the methodological decision to calculate the MAOC capacity can have implications for SOC management. The NBL method seemed to provide a better fit to our data, given the assumption that MAOC content can change across the fine fraction range, suggesting that future work

should be directed towards the NBL method. Whereas the type (mineral or organo-organo) and strength of C bonds can only be evaluated at nanoscale level with time- and cost-intensive analysis, we suggest that the NBL method can lead to a less-biased quantification of the SOC risk.

The SOC risk index showed a more refined distinction in C accrual potentials compared to considering only the MAOC saturation (Fig. 6b). Across Europe, there are a variety of locations that are under high hazard, in the sense that they are characterized by a negative ΔSOC but are below the median MAOC saturation ('HH'), spanning between 30–70 Mha, depending on the regression method. This situation occurs in particular across Scandinavia, central England, western France and some parts of the Mediterranean. The opposite combination, above the median MAOC saturation and positive ΔSOC ('NH'), occurs in the northern UK, the Massif Central (FR), as well as in Austria and southern Germany. The areal extent under no hazard, 'NH', ranges from 25–48 Mha. Locations that are at high risk, above median MAOC saturation and negative ΔSOC ('HR') cover an area of 43–83 Mha and occur mostly in countries bordering the Baltic Sea as well as northern Germany and east England. Lastly, the areas where SOC is least sensitive to losses and are at deficit in MAOC (below median MAOC saturation) are in the no risk class ('NR'). These areas could be potential locations for efficient C accrual through carbon farming. The main areas stretch from the west-coast Europe towards the east, across the countries of northern France, Belgium, and southern Germany to Hungary. Other notable locations include southwestern France and the Po valley (IT) (Fig. 6b). The total area for the 'NR' category covers 26–50 Mha. We note, however, that we have listed generic geographic patterns here and there is large variability within different pedo-climatic zones, also depending on the regression method (BL, PBL, NBL) used. The 'high hazard' and 'high risk' index classes were associated with larger uncertainty, given the larger range in their areal extent based on an uncertainty propagation analysis (Table 1).

Overall, however, there was 59.6% agreement between all three regression methods on the SOC risk index, while there was some method-dependent spatial disagreement, in central and south-west England, the Iberian peninsula and south Germany (Supplementary Fig. 18). The ratio between agreement/disagreement varied strongly between SOC risk index classes (Table 2), in particular for classes below the median MAOC saturation (NR and HH). These differences can likely be attributed to the different assumption of each regression method to estimate the effective MAOC capacity for coarse soils. Based on this 'convergence of evidence' approach, we conclude that the SOC risk index provides robust information for areas where to prioritize measures to revert degrading processes or protect the existent SOC pool (HR and NH classes), while there is more uncertainty on areas suitable and with some potential for SOC accrual (NR and HH classes).

## Table 1 | Summary table of the corresponding area (Mha) for each SOC risk index class

| SOC risk index | Method | Mean | Q5 | Q95 |
|---|---|---|---|---|
| High hazard (HH) | BL | 59.2 | 54.4 | 61.7 |
| | PBL | 30.0 | 25.9 | 33.7 |
| | NBL | 69.8 | 67.0 | 75.4 |
| No risk (NR) | BL | 46.0 | 42.6 | 48.0 |
| | PBL | 26.2 | 23.6 | 29.1 |
| | NBL | 49.5 | 47.0 | 52.3 |
| High risk (HR) | BL | 54.0 | 51.5 | 58.9 |
| | PBL | 83.2 | 79.5 | 87.3 |
| | NBL | 43.4 | 37.8 | 46.2 |
| No hazard (NH) | BL | 28.7 | 26.7 | 32.0 |
| | PBL | 48.5 | 45.5 | 51.0 |
| | NBL | 25.1 | 22.4 | 27.6 |

For details on the regression methods, see Fig. 4 in the main manuscript. The mean values in Mha correspond to the values discussed in the main text, whose spatial patterns across Europe are provided in Fig. 6b. Q5 and Q95 refer to the 5th and 95th quantiles that were calculated based on an uncertainty propagation analysis (Methods section, Supplementary Figs. 22-23).

## Table 2 | Agreement between methods to estimate effective mineral-associated organic carbon (MAOC) capacity and their estimated area for each soil organic carbon (SOC) risk index class in million hectare (Mha)

| Agreement | SOC risk index | MAOC | ΔSOC | Area (Mha) | % |
|---|---|---|---|---|---|
| Agreement | High hazard (HH) | < median | < 0 | 27.5 | 46.4 |
| | No risk (NR) | < median | > 0 | 24.6 | 53.5 |
| | High risk (HR) | > median | < 0 | 39.0 | 72.2 |
| | No hazard (NH) | > median | > 0 | 20.8 | 72.6 |
| Disagreement | High hazard (HH) | < median | < 0 | 31.7 | 53.6 |
| | No risk (NR) | < median | > 0 | 21.4 | 46.5 |
| | High risk (HR) | > median | < 0 | 15.0 | 27.8 |
| | No hazard (NH) | > median | > 0 | 7.9 | 27.4 |

Agreement means that all three methods allocate a raster cell to the same SOC risk index class. The relative percentage of agreement between the regression methods is calculated as area of agreement / total area for each SOC risk index class. Where the SOC risk index classes based on the non-linear boundary line (NBL) method have been taken as a reference for Table 2. The columns MAOC and ΔSOC indicate which quadrant of the risk index that row belongs to (Fig. 6b). For details on the regression methods, see Fig. 4 in the main manuscript.

The magnitude of saturation also determines the rate of soil C accrual[16,44], which affects the extent to which carbon farming is likely to be a cost-effective practice[45]. Conversely, the risk index is built on the concept that soils close to MAOC saturation lead to more rapid losses, as shown in a global synthesis[16], due to higher POC content and weaker MAOC sorption/bonds to the mineral matrix[11,13,44]. Our data showed that high levels of MAOC saturation also led to higher C losses on a continental scale, consistently across all three boundary line regression methods (Supplementary Fig. 19, Supplementary Table 3). We further note that suitability for soil C accrual or SOC protection measures should not only focus on the degree of saturation but also on the absolute amount (i.e. in terms of g kg$^{-1}$ of C) of MAOC. For example, locations that are under low risk and, thus, have more potential for SOC accrual ('NR'), have a range of effective carbon storage potentials, which depends on their pedo-climatic cluster associations and the boundary line regression method (Supplementary Fig. 20). While other biophysical limitations exist for SOC accrual practices, such as the availability of nutrients[46,47] and soil depth[17], our index identifies potential regions for C accrual and protection, acknowledging constraints in terms of soil characteristics, NPP, climate and land use. Socio-economic and technical constraints may also limit the adoption of farming activities that aim at accruing or protecting SOC as, for instance, access to farm advisory services or risk aversion with respect to alternative management practices[48]. Future research could focus on expanding the MAOC dataset by additional soil sampling and C fraction measurements, to cover a wider range of soils and environmental conditions. Lastly, we have shown that calculating the degree of C saturation is affected by methodological decisions and we hope that our findings lead to further research towards a unified approach.

## Methods

### Analytical data

The LUCAS dataset consists of records from a 2009 sampling campaign, based on a random sampling design stratified by land use and topography. Soil cores were taken at a depth of 0–20 cm, see Tóth et al. for further details[28]. The bulked soil samples were air-dried and sieved to their <2 mm fractions. Soil analytical data for clay, silt, organic carbon and pH in H$_2$O was determined by standard methods following ISO protocols (Supplementary Table 4). Soil spectra in the visible- and near-infrared range (VNIR, 380–2500 nanometer (nm) range) were measured with a XDS Rapid Content Analyzer (FOSS NIRSystems, Inc., Denmark) at 0.5 nm spectral resolution. The protocols of the instrument manufacturer and the soil spectroscopy group[49] where followed for the spectroscopic measurements. For each sample, the mean spectrum was taken of two replicates. We restricted the LUCAS 2009 dataset to locations that were both under agricultural land use, as recorded by the surveyors corresponding to cropland and grassland under 1000 m a.s.l[8]., and had associated VNIR spectra ($n = 13,295$).

The analytical soil C fraction data was originally measured for a selection of soil samples from the LUCAS 2009 survey[3], the procedure of which we briefly summarize here. Firstly, the aggregates were dispersed. Five grams of oven dried, <2 mm sieved soil was shaken for 18 hours in dilute (0.5%) sodium hexametaphosphate with beads. After aggregate dispersion, samples were fractionated by size through rinsing the soil samples onto a 53 μm sieve (see[3,27] for further details). Where the < 53 μm fraction was considered MAOC and the > 53 μm was considered POC[50]. We then also restricted the soil C fraction dataset to locations that were both under agricultural land use and had associated VNIR spectra ($n = 240$).

The processing of the spectra was done by sub-setting every 10$^{th}$ wavelength, trimming the spectra to the range of 400–2450 nm and computing the 1$^{st}$ derivative. Subsequently, the H$_2$O bands were removed from the spectra by excluding the 1350–1460 nm and 1790–1960 nm wavelength regions[51].

### Calibration regression

Based on the results from an exploratory analysis (Supplementary Fig. 2), we decided to use a local partial least squares regression method. For the calibration regression method, we adapted the method described in Summerauer et al. (2021) to our purpose[52]. We used the moving-window correlation as a metric to select $k$-nearest neighbors based on spectral similarity. In order to choose the window size, we computed the RMSE between nearest neighbors for different window sizes (11-151 in steps of 10). We selected the window size with the lowest RMSE (Supplementary Fig. 21). After the nearest neighbors had been selected, a local model was fitted based on the weighted average partial least squares regression algorithm as per Shenk et al. (1997)[53]. For each number of components used in the PLS, from 1 to $j$, a weight is calculated based on the spectral residuals of the observation to be predicted. These weights are then used to average multiple PLS models computed for different number of components:

$$w_j = \frac{1}{\delta_{1;j}g_j} \tag{1}$$

Where $\delta_{1;j}$ is the RMSE of the spectral residuals for a predicted sample based on $j$ PLS components, $g_j$ is the RMSE of the regression coefficients which corresponds to the $j^{th}$ PLS component (more details in ref. 53). We considered a range of 5 to 15 PLS components.

The number of $k$-nearest neighbors was optimized by using nearest neighbor cross-validation[54] (Ramirez-Lopez et al. 2013). This method is essentially equivalent to a leave-one-out approach where for $k$ nearest neighbors, each neighbor is excluded iteratively and predicted by a weighted PLS regression using the $k$-1 nearest neighbors. The predictions are then cross-validated against their analytical values. We considered a value of $k$ between 20 and 100 where the final $k$ value was selected based on the minimum RMSE in the nearest neighbor cross-validation. We then restricted the LUCAS 2009 dataset to the SOC range of the soil C fraction calibration set (3.6–85.1 g kg$^{-1}$ SOC, $n = 12,019$) and predicted MAOC and POC using the method described above.

### Determination of the calibration applicability domain

The aim of the VNIR predictions was to extend the soil C fractionation data across the entire LUCAS 2009 soil dataset. Thus, we needed to determine the applicability domain of our calibration regression based on our calibration set ($n = 240$). An established method to do this for PLS predictions is through use of the $F$-ratio[55-57]. The main idea is to assess how well the PLS scores can reproduce the spectra of the validation set compared to those in the calibration set. This is achieved by dividing the residual variance of the spectra of the validation set by those of the calibration set:

$$F = \frac{(\mathbf{u} - \hat{\mathbf{u}})^{\mathsf{T}}(\mathbf{u} - \hat{\mathbf{u}})n_c}{s_c^2} \tag{2}$$

Where $\mathbf{u}$ is the spectrum of the observation to predicted, $\hat{\mathbf{u}}$ is the spectrum of the observation to be predicted produced from the PLS scores, $n_s$ is the number of observations in the calibration set and $s_c^2$ the residual spectral variance of the calibration set. We computed the residual spectral variance of the calibration set with the projected PLS scores, whereas residual spectral variance of the validation set is computed by use of the predicted PLS scores. We then computed the probability of the $F$-ratio as per Dangal et al. (2019) and assigned a prediction as being out of the calibration applicability domain for probabilities exceeding 0.99. We then merged the POC and MAOC predictions into a single dataset, disregarding outliers for both predictions ($n = 6548$) and setting negative predictions to 0. In order to validate our predictions, we assessed our predictions against the measured SOC content. We compared the sum of POC and MAOC (Supplementary Fig. 3a) and SOC predictions directly from the VNIR

spectra (Supplementary Fig. 3b). We evaluated predictions based on the following metrics: root mean squared error (RMSE), correlation coefficient ($R^2$), bias, Lin's concordance correlation coefficient (CCC)[58] and the ratio of the standard prediction error over the inter-quartile range (RPIQ)[59].

We also visually examined the model applicability domain through the use of principal components analysis (PCA) on the VNIR spectra. We plotted the joint distribution of the first two PCA components (explaining 75.8% of the variance) to assess whether the samples within the model applicability domain lay within the range of the calibration set (Supplementary Fig. 3c-d). We note that defining the model applicability domain reduced the number of predictions by almost half. This can be partially attributed to the limited representation of the calibration dataset with respect to the LUCAS 2009 survey (Supplementary Fig. 3c). Ideally samples should span the range of spectral variability, which did not seem to be the case according to this diagnostic plot. Additionally, the spectral information might be limited in terms of the absorption features related to carbon fractions in the VNIR range. That is, previous studies have found that mid-infrared soil spectra can lead to better predictions of soil C fractions[60,61], although this likely depends on the fractionation method[62] and soil characteristics in the study[63].

## Clustering

We then clustered the LUCAS dataset ($n = 13,295$) into pedo-climatic zones based on a combination of climate, pedological and landscape factors[21-23]: i.) measured pH in $H_2O$[28], ii.) landform classes computed from digital elevation data[35], iii.) MODIS/Terra cumulative net primary productivity (NPP, kg C / $m^2$ / year)[34] and iv.) the aridity index (precipitation / potential evapotranspiration) based on the TerraClim dataset[33]. Both NPP and the aridity index were extracted using Google Earth Engine and the mean was calculated over the period 01-01-2001– 01-01-2021 at 1000 m resolution. We applied the $k$-means method using the Hartigan and Wong (1979) algorithm[64]. All variables were scaled to unit variance. We ran the following iteration 100 times for different seeds (random number generators): within each iteration the $k$-means was ran 100 times for different random allocations of initial centers. We considered a maximum of $k = 20$ to minimize over-dispersion of the LUCAS dataset, and selected the number of clusters that minimized the within-cluster sum of squares based on the elbow method, i.e. the minimum of the $2^{nd}$ derivative. From the 100 iterations, we then selected $k$ that was most frequent (see Suppl. Material for more details). These cluster associations were then used for the dataset that disregarded outliers of POC and MAOC predictions ($n = 6548$).

Additionally, we computed a random forest regression between the cluster associations and the variables used in the $k$-means method (scaled pH in $H_2O$, NPP, landform and aridity). The regression allowed us to upscale the cluster associations and thus was used to predict clusters with 1000 m grid resolution and the same extent as the Europe-wide raster dataset provided in De Rosa et al., reporting SOC changes in the period 2009–2018 ($\Delta$SOC)[8]. The raster extent corresponds to areas that were under agricultural land use (cropland or grassland), as per the Corine Land Cover dataset (https://land. copernicus.eu/pan-european/corine-land-cover). The pH in $H_2O$ and the fine fraction rasters were taken from the study by Ballabio et al.[65,66]. The raster with predicted cluster associations was used in the last step of our methodology to calculate the SOC risk index. All rasters that were not at 1000 m grid resolution (pH in $H_2O$, fine fraction, $\Delta$SOC and landform) were resampled using bilinear interpolation.

## Quantifying the degree of MAOC saturation and the SOC changes

We explored three different regression methods to estimate the effective MAOC saturation capacity as a function of the fine fraction, based on different hypotheses of MAOC saturation dynamics (Fig. 4). The first

was the boundary line regression (BL)[15]. The second, PBL, was an alternative method to filter for samples that are likely to contain POC in the MAOC fraction. This might occur during size separation, both due to POC fragmentation and dispersion of POC into dissolved organic carbon (DOC)[26,42]. We first expressed the predicted MAOC as a function of the fine fraction (g MAOC in $kg^{-1}$ fine fraction). We then determined the break-point of a piecewise linear $90^{th}$ quantile regression while constraining the slope of the second linear equation to 0, such that the effective MAOC capacity was constant across the remaining fine fraction range (as first proposed in Hassink, 1997[12]). Any values prior to the breakpoint were then considered as MAOC saturated. The breakpoint was determined through use of the segmented() function in R[67]. The third method, NBL, was a non-linear boundary line regression, also on the $90^{th}$ quantile and restricting the intercept to 0. For the non-linear quantile regression we considered a logarithmic model:

$$y = \alpha + \beta \log(x) \tag{3}$$

where $y$ = the effective MAOC capacity, $\alpha = 0$, $\beta$ is the coefficient to be estimated and $x$ is the fine fraction (clay + silt / %). For all three regression methods, the degree of MAOC saturation was calculated as (MAOC / effective MAOC capacity) × 100. Values below 100% indicate a saturation deficit relative to the cluster-dependent effective MAOC capacity. Values above 100% where considered saturated and set to 100% for subsequent analysis. This procedure was repeated for each pedo-climatic cluster.

To investigate the change in SOC ($\Delta$SOC) as a function of the ratio of MAOC to SOC (Fig. 3), we used $\Delta$SOC values obtained from the non-linear regression model developed by De Rosa et al.[8]. The trained model was used to analyze the changes in SOC between the LUCAS 2009 and 2018 surveys across the EU + UK. Since the regression model used in De Rosa et al. study depends on land use information collected over time, our analysis was constrained to sites that had repeated recordings of land use across surveys. As a result our dataset of predicted MAOC and POC was reduced to 5482 points for our investigation of $\Delta$SOC as a function of MAOC:SOC (Fig. 3).

## The SOC risk index

Lastly, we mapped at 1000 m resolution the degree of MAOC saturation and predicted $\Delta$SOC to assess the vulnerability and hazard of exposed SOC in agricultural lands. We considered the level of risk for SOC as a function of both vulnerability and the hazard. This concept follows the framework introduced by the Intergovernmental Panel on Climate Change[31] which we have adapted to our purpose (Fig. 1). The degree of MAOC saturation was taken as a measure of vulnerability, given that soils saturated in MAOC are more likely to either have a high SOC content and/or have proportionally higher POC. The $\Delta$SOC was considered as the level of hazard, that is, SOC changes driven by climatic conditions and land use change[8]. We transferred the relationship between the degree of MAOC saturation and the fine fraction content based on the cluster associations. We did this by calculating the mean degree of MAOC saturation across fine fraction bins of 10%. We then mapped the mean MAOC saturation by fine fraction bin (Fig. 5) to a raster of the pedo-climatic clusters (Supplementary Fig. 17a) and the fine fraction raster[65] that was reclassified to align with the fine fraction bins. In a few cases, the range of fine fraction bins of the data (Fig. 5) did not cover those present in the raster. In that case, we considered the mean degree of MAOC saturation to be missing and those locations were thus excluded from the subsequent analysis. Finally, the SOC risk index was calculated across regression methods (BL, PBL, NBL) by determining for each raster cell whether: i.) it was above- or below the median degree of MAOC saturation across Europe, ii.) the $\Delta$SOC was below 0 or equal to 0 and above. We thus ended up with the following four classes: 'HR': at risk, above the median MAOC saturation and negative $\Delta$SOC. 'HH': high hazard, negative $\Delta$SOC and below the

median MAOC saturation. 'NR': low risk, positive ΔSOC and below the median MAOC saturation. 'NH': positive ΔSOC and above the median MAOC saturation. Although the median degree of MAOC saturation across the three regression methods (68%) is arbitrary, there is no scientific consensus yet on a generic threshold of MAOC saturation where C accrual diminishes and a C losses become more likely. We note that the linear model fitted on the ΔSOC and the MAOC saturation (Supplementary Fig. 19, Supplementary Table 3) supports the decision to take 68% as a threshold. Given the fitted parameters, the MAOC saturation before ΔSOC goes negative is 59%, 68% and 71%, for the BL, PBL and NBL method, respectively.

### Uncertainty propagation

We have performed an uncertainty propagation analysis based on the associated error with the MAOC predictions from the soil VNIR spectra. To assess the effect of marginal uncertainties in our MAOC predictions, we have approximated the expected error based on the predicted POC + MAOC vs. measured SOC (Supplementary Fig. 3). Given the negatively skewed distribution of SOC, we calculated the mean absolute log error (MALE). The MALE is robust to outliers (high SOC values). MALE reduces the effect of large differences between the predicted and measured values and provides a better measure of the relative difference. That is, the exponential of the MALE (EMALE) represents the relative multiplicative error (once we subtracted 1). We assumed the error to be normally distributed around the mean prediction and that POC and MAOC contribute equally, so we divided the EMALE by two. We then performed 500 simulations where we resampled the mean MAOC prediction with a standard deviation represented by (EMALE-1) x MAOC (Supplementary Fig. 22). We then calculated the MAOC saturation and SOC index for each of these 500 realizations of MAOC. We calculated the 5th and 95th quantile for the MAOC saturation (Supplementary Fig. 23) and for the areas of each SOC index class (Table 1). See the Supplementary Material for further details.

Data handling, analysis and visualization was done using the following R packages: data handling with **tidyverse**[68], **prospectr**[69], **broom**[70], regressions with **quantreg**[71], **pls**[72], **caret**[73], **resemble**[74], **mgcv**[75], **mgcViz**[76], **segmented**[67], **emmeans**[77] handling of spatial objects using the **raster**[78] and **sf**[79] packages, clustering with the **cluster**[80] package. Graphics were created with base R functions, **rnaturalearth**[81], **patchwork**[82] and the package **ggplot2**[83].

### Data availability

The LUCAS 2009 soil survey and SOC fractionation data used in this study are available in the European Soil Data Centre (ESDAC) of the European Commission – Joint Research Centre under: http://esdac.jrc.ec.europa.eu/content/lucas-2009-topsoil-data and https://esdac.jrc.ec.europa.eu/content/soil-organic-matter-som-fractions. The main outputs of this study are available at: https://esdac.jrc.ec.europa.eu/content/soil-carbon-risk-index.

### Code availability

The most relevant R scripts to this manuscript are available at: https://esdac.jrc.ec.europa.eu/content/soil-carbon-risk-index.

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

## Acknowledgements

The LUCAS Soil sample collection is supported by EUROSTAT and the following Directorate-Generals of the European Commission: Environment (DG-ENV), Agriculture and Rural Development (DG-AGRI) and Climate Action (DG-CLIMA). We thank Beatrice Landoni and Christopher Havenga for their help with figure formatting.

## Author contributions

T.S.B. led the study's design, data analyses, interpretations and writing; D.D.R. contributed to the study's design, data analysis, interpretations and writing; P.P. contributed to interpretations and writing; M.F.C. contributed to interpretations and writing; A.J. conceived the funding; E.L. led the study's design and contributed to data analysis, interpretation and writing.

## Competing interests

The authors declare no competing interests
