## [Peer Review file · Nature Communications]

Revisiting the soil carbon saturation concept to inform a risk index in European agricultural soils

Corresponding Author: Dr Timo Breure

Version 0:

Reviewer comments:

Reviewer #1

(Remarks to the Author)

Thank you for the interesting paper on MAOC saturation and the proposed risk index.

The novelty of the paper seems to lie in the derivation of the risk index, although I am not entirely convinced about its derivation or how what additional benefit this provides over, for example, a map of C deficit or C sequestration potential.

There is much interest in the MAOC saturation concept currently and several recent papers have investigated the concepts and the various methods used to derive it (Hassink, Six, Feng, Giorgieou, and ViscarraRosel, others).

The idea that the maximum capacity inferred from the fine-fraction only isn't all that useful has also been previously proposed - This of course makes sense and it is well understood that the effective, or attainable MAOC capacity is what one needs, ie not only the fine fraction but also the soil conditions, climate, etc (ie pedoclimatic zones) - see for example ViscarraRosel 2023.

The regression methods proposed for estimating the capacity seem not to fit the data well - I do not see the advantage of the Msat over the original linear regression proposed by Hassink or the quantile regression approach - in fact, it seems that the fit will underestimate more than the Hassink/quantile method. The Wsat method also does not fit the data well - in fact, this is possibly worse as there will be significant overestimations when the proportion of the fine fraction is small and large underestimation when the proportion is large. The authors should use the frontier-line method proposed by ViscarraRosel 2023, to 'envelope' the maxima and prevent under and overestimations.

Regarding the derivation of the pedoclimatic zones, this makes sense, of course but I wonder if the authors tried to more simply use soil type? Soil types represent pedoclimatic regions - as Jenny's cl,o,r,p,t model shows. This might reduce complexity and be more intuitive and useful.

I found your paper a little bit 'too complicated' what is it about? the Msat and Wsat? is the spectroscopy a novelty too? or is it the derivation of the pedoclimatic regions? - the title seems to suggest that it is the risk index, but there is little on that. Perhaps you propose that it is about all of those - I would suggest that you reduce methodological complexity and focus your study on the most important aspect.

(Remarks on code availability)

Data availability is there. However, I could not see the code availability statement.

Reviewer #2

(Remarks to the Author)

Breure et al. revisited the soil organic carbon (SOC) saturation concept and developed a novel risk index for soil carbon management in European agricultural soils. This significant work leverages a comprehensive soil information database (LUCAS) and integrates the authors' past research on soil carbon fractions and trends (Cotrufo et al. 2019; Lugato et al. 2021; De Rosa et al. 2023). They introduce novel methods for characterizing mineral-associated organic carbon (MAOC)

saturation using nonlinear regression and pedo-climate clustering. Their risk index is particularly intriguing as it incorporates both MAOC saturation and SOC trends in an elegant fashion. Their work and risk index have the potential to be highly influential in guiding SOC farming and management efforts.

Overall, this manuscript is technically sound and well written. It represents the forefront of the current research and is suitable for the audience of Nature Communications. I have some concerns regarding the representativeness of their SOC fraction dataset. The authors might consider acknowledging limitations of their SOC fraction dataset and discuss the implications and future directions to further enhance the applicability of their findings.

General comments:

I have concerns about the size and potential bias of the training dataset used for SOC fractions, initially reported by Lugato et al. (2021) and comprising 240 samples. This sample size is significantly smaller than that used to determine MAOC behavior ($n = 6548$, as shown in Extended Data Fig. 1). The limited sample size presents several issues:

The SOC fraction dataset lacks samples with high fine fraction and high MAOC content, which likely affects the modeling of MAOC saturation. Throughout their entire dataset, the MAOC content never exceeds 50 g kg^{-1} (Extended Data Figs. 3 and 6). This limitation likely stems from the absence of high MAOC soils in their SOC fraction data, as their random forest model would not predict MAOC levels beyond the maximum observed in the training data. A recent study by Begill et al. (2023) has identified the presence of soils with high MAOC and high fine fraction in European agricultural systems Europe. Missing these soils could lead to an underestimation of the effective MAOC capacity, affecting the model's accuracy regardless of the modeling approach, and potentially skewing the estimation of risk factors. 2) The small sample size may also contribute to the relatively low predictive power of the model in estimating MAOC (Extended Data Fig. 2; Line 127 to 130). The data cloud distribution for the measured and predicted variables is very wide. The VNIR spectra do not appear to be good predictors of C fractions and SOC content. Given this sample size, it is challenging to assess whether the MAOC model is effective across different pedo-climate clusters.

I recommend that the authors address the potential limitations and implications associated with the size of their SOC fraction dataset. While their research undoubtedly represents the forefront of the discipline, discussing these limitations will not only acknowledge the current dataset's potential constraints but also provide valuable insights for advancing future research in this area.

Specific comments:

Beginning at line 130, but generally concerning the methodology for ΔSOC . Were the changes in SOC estimated or were they measured? My impression from reading the De Rosa et al. (2023) is that the ΔSOC were calculated using paired SOC measurements. However, ΔSOC was frequently referred to as estimated or predicted (e.g., line 130, 148). It would be helpful to specifically state how this parameter was calculated and clarify the methodology.

Beginning line 193, it is evident that the W_{sat} provided a better fit for the predicted MAOC data compared to H_{sat} . This non-linear model also predicts lower effective MAOC capacity than those reported in past studies (Feng et al. 2013; Georgiou et al. 2022; line 216 to 221). These results could reflect the data pattern where the predicted MAOC remain relatively flat in soils with high fine fractions (Extended Data Figure 4). Could this pattern be associated with the lack of soils with high fine fractions and high MAOC? I agree with the authors that the LUCAS sampling design is sufficient to capture the most abundant soil types in European (line 222). However, there remains the concern that their SOC fraction samples might not be a good representation of European soils. If there is an artificial threshold due to the limited sample size, then is it fair to conclude that W_{sat} is an effective method for estimating effective MAOC capacity?

Beginning at line 300, but for the entire SOC risk index paragraph. A table for the risk index categories & their combination of MAOC saturation and change in SOC may be beneficial for future readers.

Line 294, I wonder why the median MAOC saturation was chosen as the classification criterion.

Line 306, it might not be fair to refer to these soils as "MAOC saturated". They are more MAOC saturated than those in HE and LR, while they might not approach 100% saturation.

Line 317, the similarity observed between three regression methods could also be explained by the use of median MAOC saturation as the classification criterion. In each cluster, while the regression methods influence the estimated degree of saturation in percentage, they do not affect the placement of a sample within the distribution of other data points. Thus, a soil will be categorized into either more saturated or less saturated group regardless of the regression methods.

I wondered whether the authors have plotted the ΔSOC against the estimated degree of saturation? According to the theory behind MAOC saturation, soils with lower MAOC saturation would likely gain SOC more rapidly than more saturated soils (Georgiou et al. 2022). Similarly, more MAOC saturated soils might lose SOC more quickly. I think the authors have a rare opportunity to explore these hypotheses using a comprehensive regional dataset.

(Remarks on code availability)

Reviewer #3

(Remarks to the Author)

(Remarks on code availability)

Version 1:

Reviewer comments:

Reviewer #1

(Remarks to the Author)

(Remarks on code availability)

Reviewer #2

(Remarks to the Author)

I appreciate the authors' response and revision in response to my previous comments. They included new text on the potential limitation of their methods and the future research directions. Their clarification about some methods and the recalculation of the risk index are also welcome. I have some follow-up regarding the authors' responses to my comments and the next text:

Regarding comment 6 from the previous review, the authors' responses make sense, and their arguments are well laid out in the methods section (L493-L503). Personally, the sample size of the fractioned data is still small ($n=240$) for a large-scale study covering Europe. Compared to my other comments (6, 8, 9, and 11), my main concern was about the sample size and potential consequences. Thus, I would suggest that the authors acknowledge it in the main text.

Regarding the comment 7, I do not agree with the statement that high MOAC soils in Begill et al. (2023) are mostly Gleysols and Stagnosols. My group re-analyzed their data, which are publicly available, and yes, their dataset does include Gleysols and Stagnosols, indicative of hydric conditions that contribute to OM accumulation that are distinct from the hypothesis of MAOC accumulation used in the POC/MAOC framework. However, high MAOC content soils represented in that dataset are not exclusive to the Gleysols and Stagnosols. Six et al.'s reanalysis also supports this assessment (DOI: 10.5194/soil-10-275-2024). Furthermore, the global dataset from Georgiou et al. 2022 contained many data points with MAOC above 50 g C kg⁻¹. Thus, there is still the concern that the maximum MOAC content in LUCAS (Line 250) might be an artifact due to sampling design and sample size. In sum, the issue is bigger than the soils with oxygen limitations (as portrayed by the authors L260-264).

I appreciate the renaming of the four categories for the risk index, as they are more intuitive than the last version. I still have some trouble differentiating High Risk from High Hazard. In fact, it is hard to remember which is which. In the main text, could the authors elaborate on where the names came from?

Regarding the table about SOC risk index, moving it from the supplement to the main text is a good starting point but does not address the full extent of the original comment. The main idea behind the inclusion of this table is to make this concept as intuitive and understandable as possible for future readers. There are a few issues with the table: 1) there is enough space to expand the abbreviation of column two to include the full name of the risk category; 2) one of the abbreviations should be "HN" rather than "LH"; 3) Perhaps change the title of the third column from Mha to Area (Mha); 4) It may be helpful to show the combination of change in SOC and MAOC saturation that corresponds to each category. For example the positive change, no change, and negative change in SOC described in lines 325-326, as well as above and below MAOC saturation in lines 323-325

(Remarks on code availability)

The codes were mostly for a published paper. I cannot find codes specific to this study.

Reviewer #3

(Remarks to the Author)

(Remarks on code availability)

Version 2:

Reviewer comments:

Reviewer #2

(Remarks to the Author)

Thanks to the authors for considering my comments and addressing them. Most of comments have been satisfactorily resolved. However, I still have concerns about Table 1. The High Hazard and No Risk groups appear to be mislabeled. According to L332-333, High Hazard is associated with SOC losses and MAOC saturation below median. In contrast, Table 1 shows that the High Hazard group is characterized by SOC gains. Additionally, some of the area estimates seem inconsistent. The text (L345-346) states that High Risk group covers an area of 43-83 Mha; however, these numbers do not match with those from the table ($27.5+2.5 = 30$ Mha). Given the importance of the risk index in the authors' novelty arguments, caution should be exercised to ensure the accuracy of this table and associated results, including Figure 5 and related supplementary figures.

(Remarks on code availability)

According to the authors, codes will be made available after acceptance. I could not review their codes at this stage.

Reviewer #3

(Remarks to the Author)

"I co-reviewed this manuscript with one of the reviewers who provided the listed reports. This is part of the Nature Communications initiative to facilitate training in peer review and to provide appropriate recognition for Early Career Researchers who co-review manuscripts."

(Remarks on code availability)

Reviewer #4

(Remarks to the Author)

The authors propose a risk index as a combination of vulnerability (MAOC saturation) and hazard (Δ SOC) for agricultural soils. The novelty lies mainly in the methodology of estimating MAOC saturation. The authors' idea is based on the effective "biophysically achievable" MAOC capacity determined from clustering pedo-climatic conditions across Europe and using three different regression methods to estimate saturation separately for each cluster.

The problem of defining realistic carbon sequestration goals and focusing efforts on most promising areas is globally relevant, and the discussion of different methodologies to address this problem contributes to both science and policy development. From a policy perspective, the resulting index mapped over the EU could be considered the main contribution. From the science perspective the paper offers two main opportunities: 1) discussion of MAOC capacity as an emergent ecosystem property; 2) discussion of regression methods used to estimate capacity from observations. In my opinion, both topics are not fully addressed in the current manuscript and can be improved upon as detailed below.

1) The authors argue that the effective MAOC capacity cannot be determined solely based on the clay content. Previous studies shared the same argument and added clay mineralogy (Georgiou et al., 2022) and soil types (Viscarra-Rossel et al., 2023) to MAOC capacity estimation. Here the authors chose to stratify soils by pedo-climatic clusters based on pH, landform class, NPP, and aridity index. I would like to see more discussion of this choice of factors in terms of:

a) Justification of factor selection. Adding or removing a factor can significantly change clustering results, therefore the readers should be able to see the logic of e.g. including landforms, or e.g. not including nutrients, total SOC or plant functional types.

b) Even more importantly: understanding the mechanisms behind the clusters. It is implied that the clay-MAOC relationship is modified by the pedoclimatic conditions, represented by the 4 abovementioned factors. But there is no discussion (or better quantitative analysis) of why that would be the case (and why the selected factors are deemed more important than the omitted ones).

c) While effects of the 4 pedo-climatic factors could be initially hypothesized, eventually the authors could analyze their results to see the individual effects of each clustering factor on the estimated effective MAOC capacity. Does low pH tend to increase or decrease it? Why? How are landforms affecting MAOC capacity? Could they be a proxy of mineralogy, or water redistribution, or erosion? Such analysis and discussion may greatly improve the scientific insight that the reader could draw from the paper, compared to a probably correct, but rather obvious statement that "clay alone is not enough".

d) The authors' definition of effective MAOC capacity differs from Georgiou et al. (2022) and Viscarra-Rossel et al. (2023) not only in the selection of factors, but in a more fundamental way. While previous definitions focused on practically unchanging properties, i.e. texture, mineralogy and soil types, in this paper the estimation is based on dynamic properties i.e. NPP, pH and aridity. Thus, in the previous studies, "MAOC capacity" was an invariant physical soil property, to which actual MAOC

content could be compared, and a “stationary target” which land users may strive to achieve through improving their practices. In this paper, the authors use the same “MAOC capacity” term for an ecosystem property that is altered by climate change (aridity index, NPP), and land use (NPP and pH). I understand that authors used 20-year averages for aridity and NPP counting it as long-term climate. But let’s imagine we use their MAOC capacity as a target for a long-term land management project today. The result of the project may be evaluated in 20 years - a reasonable time for changes in slow-turnover carbon pools such as MAOM, and quite a significant time considering rapid climate change and even faster land management changes. By then we will have different aridity, different NPP, and possibly different pH, especially if there was a substantial change in land use practices. So, the same land area may get into a different cluster in 20 years, and its estimated MAOM capacity will change. We get a “moving target” instead of a stationary one. The situation may be particularly confusing in case our land management intervention is designed to increase SOC by enhancing C input to soils via larger NPP, e.g. by implementing cover crops and irrigation. In this case we will be by definition changing the MAOC capacity, rather than trying to reach it! I see the benefit of considering the whole ecosystem in realistic target-setting, but I’d like more clarity in definitions here. E.g. we should avoid the confusion of using the same terms “MAOC capacity” for a time-invariant soil physical characteristic and simultaneously for a time-dependent ecosystem property. The authors use the term “effective MAOC capacity”, which I think should be more clearly defined, especially in the context of the methodology potentially being used for agricultural land risk-assessment or climate target setting.

2) I appreciate the authors’ effort to investigate three different regression methods of estimating the effective MAOC capacity. However, it is rather unclear to me what the conclusion regarding these methods is. If anybody would like to replicate the proposed methodology for a different geographic region, would they need to re-examine all three methods? Or is there some insight on a preferable regression method that the authors would like to share based on their investigation?

3) Additionally, I would like to note that the whole calculation of MAOC saturation relies on the initial step of predicting MAOC from a limited number of measurements using VNIR spectroscopy data. I assume such prediction had a significant uncertainty and I wonder if that uncertainty was taken into account in the subsequent steps of MAOC capacity estimation and risk assessment. If it was not taken into account, then this should be done and presented in a quantitative way, as well as discussed.

4) It was not possible to completely follow the responses to previous review that was available, as no prior version was supplied, and would be beyond the scope of such a review. However, I agree with referee 1 that the manuscript even in this revised version is rather opaque. Hopefully the comments above help in a thorough re-writing that clarify the arguments made.

(Remarks on code availability)

n/a

Reviewer #5

(Remarks to the Author)

(Remarks on code availability)

Version 3:

Reviewer comments:

Reviewer #4

(Remarks to the Author)

Thank you for the revisions.

I focus on these comments on the changes made in response to my earlier comments, and any new writing.

I note that while the revisions are attempting to address the concerns and the responses are addressing much of the suggestions, the reader does not fully benefit as they are not fully implemented in the manuscript (a typical response is (not verbatim): yes, this is a challenge, but here is a reason why it is not a big problem and now we ignore it). I also note that inaccurate language creeps in, which would be possible to address and should ideally not be present at this stage of review&revision,- which makes it a bit challenging looking forward to an eventually near-flawless manuscript. I give the authors benefit of the doubt and hope for their own sake that they do not take the revisions lightly (which I think they did not do in the rebuttal, but seemed to not have taken to heart in the revised manuscript).

Detailed Comments

Line 202: what is a “organo-organo C bonds on the clay surface”? Either it is an organo-organic C bond or it is an organo-mineral bond, but a “organo-organo C bonds on the clay surface” seems a contradiction,- or they mean a organo-organic bond or OC on a clay surface. Please scrutinize your phrasing and make sure it is not confusing or inaccurate. (also L 248)

Line 203: “particularly for data low in fine fraction content” does also not make sense without some modification. How can data be low in fine fraction? Do you mean a dataset of soils are low in fine fractions? Or something else? Unfortunately a few of these inaccurate remarks are creeping in and I hope you or your co-authors can rectify it.

Line 246: “Differences in the PBL method” should probably read “Differences between the PBL compared to the BL method”.

Line 279: "We anticipate that these limitations might affect the estimate of the maximum MAOC capacity for fine-textured soils in some pedo-climatic clusters. Nonetheless, the mean": I appreciate that the authors followed up with some error calculations, but then they do not seem carry that over to the manuscript, and do not give quantitative information about how reliable the estimates are, but rather argue "...align very closely with..." without giving the reader the benefit of their estimates, but leave this referee ignored in outlining the error range for in this case the maximum MAOC; there is some guidance in the replies of the authors that would be helpful to also see in the published manuscript. If that could be done in lucid language this would be wonderful.

Lin 300: Rephrase to "soils low in fine fraction content." (data cannot be low in fine fraction)

Line 302: How many did exceed 100% in BL and NBL? Give numbers in this sentence.

Line 315: I guess that you introduce the term and concept "effective MAOC capacity" here? At least, I was not able to find the term earlier in the manuscript. What is that? (referring to the methods is not sufficient) How is it calculated? It is really good if you can write the text in a way that it can be understood without extensive supplementary reading. For example, if this is a new term in the literature or you define it in a new way, then you would want to choose words such as "we establish"; if it has not been used before in this manuscript, then at least you may want to say "we calculated"; what you write makes the reader think that this is a well known term or has been mentioned before in the text, or both,- which does not seem to be the case. (the same applies to Δ SOC in Line 349, which was also not mentioned before) This may require some significant restructuring.

Line 320: I appreciated the strong methodological pitch in this paragraph, but it did not contribute to readability,- rather it made this reader wonder whether there is anything to learn about SOC beyond differences in computational approaches. This seems to be somewhat an issue throughout: not very clear in terms of storytelling that is buried in methodological jargon. That can be fixed, and should be fixed for an interdisciplinary journal with a broad audience.

Line 332: Capitalize "findings"

The term "SOC Risk index" (Line 309) would benefit from a more intuitive description in lines 310-319 to facilitate understanding.

I hope you find these suggestions useful to make your paper accessible, and look forward to seeing it in print.

(Remarks on code availability)

Reviewer #5

(Remarks to the Author)

(Remarks on code availability)

Reviewer's comments and responses

Reviewer #1 (Remarks to the Author)

1. Thank you for the interesting paper on MAOC saturation and the proposed risk index.

The novelty of the paper seems to lie in the derivation of the risk index, although I am not entirely convinced about its derivation or how what additional benefit this provides over, for example, a map of C deficit or C sequestration potential.

There is much interest in the MAOC saturation concept currently and several recent papers have investigated the concepts and the various methods used to derive it (Hassink, Six, Feng, Georgiou, and Viscarra-Rossel, others).

The idea that the maximum capacity inferred from the fine-fraction only isn't all that useful has also been previously proposed - This of course makes sense and it is well understood that the effective, or attainable MAOC capacity is what one needs, ie not only the fine fraction but also the soil conditions, climate, etc (ie pedoclimatic zones) - see for example Viscarra-Rossel 2023.

We thank the reviewer for the comments. We agree that the risk index is an important novelty of our work, and we are sorry if we introduced it rather abruptly in the submitted version of our manuscript, without making its benefits sufficiently clear. To address this concern, we have added a rationale to explain its novelty and benefit in the text (L88-L89, L294-L300), and to further aid in its interpretation (L349 – L367).

Below, we provide some further considerations about our methodological approach to address the reviewer's concerns above:

- 1) The papers mentioned by the reviewer all estimate saturation on the basis of silt and clay, with Georgiou et al., dividing soils in two classes on the basis of clay minerals activity, and Viscarra-Rossel et al. in multiple groups on the basis of soil types. We add our own contribution to the topic of calculating the effective MAOC capacity by considering it as an emergent property from pedo-climatic conditions. As reported above, in our revision, we have however made clear how the idea stemmed from these previous analyses (L104-106, L203-L205). Not changed since first version: L61 – L68.
- 2) We indeed think that our study has the benefit of comparing different methods with the largest harmonized C fraction dataset in the EU. This allows going beyond theoretical discussions about saturation, while presenting quantitative differences of different saturation calculation approaches over a continent.
- 3) Our idea was that effective MAOC capacity only provides partial information on the SOC status. For instance, in a scenario where two locations have the same SOC changes but a different MAOC saturation level, the consequence in terms of SOC quantity and quality is expected to be different. For instance, soil with low MAOC saturation and increasing SOC could be considered not being at risk. Therefore, the risk index integrates both the

vulnerability and hazard dimension, giving a synthetic indication that may not be captured by the C deficit presented in previous works (see also comment #4 for details).

- 4) Overall, we believe, our risk index is an advancement from the previous papers that rev. mentioned, as enable identifying areas where to prioritize interventions.

Regarding the derivation of risk index, indeed, we have harmonized the threshold of MAOC saturation to facilitate better comparison between the regression methods, and we refer to comment #15 for more details.

2. The regression methods proposed for estimating the capacity seem not to fit the data well - I do not see the advantage of the Msat over the original linear regression proposed by Hassink or the quantile regression approach - in fact, it seems that the fit will underestimate more than the Hassink/quantile method. The Wsat method also does not fit the data well - in fact, this is possibly worse as there will be significant overestimations when the proportion of the fine fraction is small and large underestimation when the proportion is large. The authors should use the frontier-line method proposed by Viscarra-Rossel 2023, to 'envelope' the maxima and prevent under and overestimations.

We appreciate the Viscarra-Rossel approach (published after we performed this work). However, we write here why we think our three methods are suitable for our purpose:

1. We would like to emphasize that the rationale behind our approaches, as explained in Fig.1 below (Fig. 3 of the manuscript), underlies two different theories on OC physical saturation to the mineral matrix:
 - Hsat and Msat assume a max MAOC concentration in the fine fraction, governed by purely-mineral-organic interaction. (right-y axis in red and horizontal red line).
 - The Wsat (in analogy to Viscarra-Rossel et al. 2023) underlies a texture-dependent MAOC concentration related to likely organo-to-organo bounds in low silt+clay regions. I.e. the red line is not constant across the x-axis range (fine fraction content).

Figure 1 - Different regression methods to estimate MAOC capacity, presented as Fig. 3 in the main text.

- Since we are not measuring directly or instrumentally a physical property, any method to fit data is dependent on some assumptions, giving more or less weight to possible associated errors in the data (e.g. the choice of the quantile is always arbitrary). We aligned our choice of quantile (90th quantile) with previous publications (Feng et al. 2013 and Six et al. 2023) and compare our parameter estimates against previous studies. An assumption from the envelope method is that all high concentration MAOC points are accurately estimated. In that sense, the envelope method differs from the quantile regression, where the upper quantile range serves as a conservative estimate of the boundary, given the associated marginal uncertainties with soil MAOC analytical measurements/estimates.

This phenomenon can be seen in an example of our fitting for similar soil types (Podzols) in Australia (a.) and our study (b. and c.). Comparing the panels in Figure 2, it can be noted that the frontier-line method fits exactly on the upper boundary (although in the Viscarra-Rossel et al. 2023 there is a slightly lower estimate due to bootstrapping).

Figure 2 - Regression methods to estimate effective MAOC capacity under the assumption of non-constant maximum in the fine fraction. a.) taken from Fig. XX in Viscarra-Rossel et al. (2023), b.) original fit of cluster 5 (solid line) together with the envelope estimate (dashed line). Cluster 5 is characterized by low pH, and high rainfall/ET0 (high aridity index) aligning with the main characteristics of podzols, c.) Cluster 3 with the original fit (solid line) and the envelope estimate (dashed line).

- We agree that the original Hassink approach (Hsat) does not fit our data well (Extended Data Fig. 4 and 7). Indeed, visual inspection of the scatterplots would indicate it is mostly applicable to clusters 1, 3, 4, 8, 12 and 13. However, other studies have found that this relationship does not hold well for fine-fraction soils (Begill et al. 2023, Fig.3a). We have now emphasized this specifically in the manuscript, see L209 –L211 together with a reference to Begill et al. (2023). The Msat and Wsat methods attempt to correct this, which from our viewpoint fit the data better. Also, the model parameter estimates are comparable with previous studies, see Extended Data Fig. 4-7 and discussion section in text (L227-L245).

4. The fact that the frontier-line method is non-parametric complicates comparisons with previous studies.

Given these considerations, we have decided to include the envelop analysis and report the results in Supplementary Figure 3, which is referred to in the main text on L216-L226. This allows us to align closely to the most recent literature, but also to get a likely upper estimate of the effective MAOC capacity with our data (given the characteristics of the envelope method to fit exactly on the high concentration MAOC samples, as we illustrated above).

3. Regarding the derivation of the pedoclimatic zones, this makes sense, of course but I wonder if the authors tried to more simply use soil type? Soil types represent pedoclimatic regions - as Jenny's cl,o,r,p,t model shows. This might reduce complexity and be more intuitive and useful.

We agree with reviewer 1 that reducing complexity and the use of intuitive soil classes would be useful. Therefore, we have looked at the availability of EU soil type data based on the World Reference Base (WRB) soils from the FAO (FAO, 2023). The map available appeared relatively coarse, potentially missing spatially refined information on specific soil/soil-forming conditions. This is illustrated by the large spread of the aridity, landform, NPP and pH properties within the WRB classes. For example, the more common soil type classes (CMdy, CMeu) have a range of 4-8 for pH in H₂O (on a log-scale).

Figure 3 - World reference base (WRB) soil classes (FAO, 2022). a.) spatial distribution for the LUCAS soil data that have VNIR measurements. b.) boxplots of the cluster variables by WRB class.

We point out that one of main intentions of this work is to test whether effective MAOC capacity is an emergent ecosystem property of pedo-climatic conditions. Therefore, we used measured variables (pH), and those derived from earth observation (landform, NPP, aridity). This method of

spatial classification could be more reliable than solely extracting information from a relatively coarse soil type map, which could introduce higher uncertainty due to spatial interpolation.

To emphasize its rationale, coherence and logic, we have re-written the section on clustering. Specifically, we have moved the methodological details on transferring the clusters on the EU raster (L298—L300) and refer to the supplementary material together with the, Suppl. Figures (x 5 and 6) for the interested reader.

4. I found your paper a little bit 'too complicated' what is it about? the Msat and Wsat? is the spectroscopy a novelty too? or is it the derivation of the pedoclimatic regions? - the title seems to suggest that it is the risk index, but there is little on that. Perhaps you propose that it is about all of those - I would suggest that you reduce methodological complexity and focus your study on the most important aspect.

We agree that this is a paper dense of concepts. Indeed, using VNIR spectroscopy to extend the C fraction data across the LUCAS survey is novel and allows us to investigate our main aims:

We wanted to challenge and make evident the choice of the regression method to estimate MAOC saturation that is, in turn, the vulnerability dimension of our risk index. Since other papers have already provided spatial estimates of MAOC saturation for the EU (e.g. Georgiou et al., 2022) we wanted to add novelty to the discussion by considering the hazard component (Δ SOC). Given that both climate change and management are exacerbating SOC losses, we thought a risk index may suggest areas where to prioritize interventions. This idea built on the vulnerability-exposure-hazard risk concept from the IPCC (Figure 5 below), where: *exposure* dimension represent the SOC pool in the agricultural area; *vulnerability* equates MAOC saturation; and *hazard* the Δ SOC as the integrated effect of climate and management to which all agricultural SOC pools are exposed. We have now added this adapted framework as Extended Data Fig. 2 in the manuscript to contextualize our decision, also in line with text revisions made for comment #1.

Figure 4 - Risk framework as proposed by the Intergovernmental Panel on Climate Change. Fig. 1.5 of the sixth assessment report (Begum et al. 2022).

We improved the rationale and the interpretation of the risk index in the main manuscript (L88-89, L294-L300, L571-L573) and made amendments throughout the manuscript in order to improve readability (L308, L327, L570-577).

Reviewer #1 (Remarks on code availability):

5. Data availability is there. However, I could not see the code availability statement.

We appreciate the call for open research. Indeed, in agreement with the senior editor, we have agreed to add our code to the following repository: <https://doi.org/10.5281/zenodo.8227702>. This link contains code from previous publications (Lugato et al. 2021, Cotrufo et al. 2023) using the same dataset and/or considering the same topic of C fractions in soils. We will add it on that repository, in case the manuscript is accepted for publication.

Reviewer #2 (Remarks to the Author):

Breure et al. revisited the soil organic carbon (SOC) saturation concept and developed a novel risk index for soil carbon management in European agricultural soils. This significant work leverages a comprehensive soil information database (LUCAS) and integrates the authors' past research on

soil carbon fractions and trends (Cotrufo et al. 2019; Lugato et al. 2021; De Rosa et al. 2023). They introduce novel methods for characterizing mineral-associated organic carbon (MAOC) saturation using nonlinear regression and pedo-climate clustering. Their risk index is particularly intriguing as it incorporates both MAOC saturation and SOC trends in an elegant fashion. Their work and risk index have the potential to be highly influential in guiding SOC farming and management efforts.

Overall, this manuscript is technically sound and well written. It represents the forefront of the current research and is suitable for the audience of Nature Communications. I have some concerns regarding the representativeness of their SOC fraction dataset. The authors might consider acknowledging limitations of their SOC fraction dataset and discuss the implications and future directions to further enhance the applicability of their findings.

We thank the reviewer for this comment appreciating the value of our work. We have added different sections to better acknowledge the limitation of our dataset as reported below in response to the specific comments.

Reviewer #2 - General comments:

6. I have concerns about the size and potential bias of the training dataset used for SOC fractions, initially reported by Lugato et al. (2021) and comprising 240 samples. This sample size is significantly smaller than that used to determine MAOC behavior ($n = 6548$, as shown in Extended Data Fig. 1). The limited sample size presents several issues:

Whereas the number of samples with C fraction data for agricultural soils seems limited, we would like to contextualize it as follows:

- 1) within Europe, this dataset is still one of the largest standardized dataset with soil carbon fractions in agricultural soils. Previous studies used datasets that were composed of data from a selection of existing studies, which imposes other limitations. For example, compatibility between sample collection (e.g. depth), processing and analysis (e.g. operating procedures for analytical methods).
- 2) Additionally, the subset for carbon fraction analysis are well distributed across texture and SOC (Extended Data Fig. 1 in: Lugato et al. 2021). The LUCAS survey has been designed on purpose to be representative of European soils and their land use, using a stratified sampling scheme (Tóth et al. 2013; Orgiazzi et al. 2018), see also L401-L404.
- 3) Given the limited number of calibration samples, we applied a strict methodology to determine the model applicability domain (Ext. Data Fig. 2) based on published methodologies (Marten and Naes, 1989; Leifeld, 2006; Dangal et al. 2019). Indeed, we found that 6,548 samples lay within the model applicability domain (L480-L481). As a diagnostic, Ext. Data Fig. 2d show that this restricted dataset fits well within the range of VNIR data, based on the joint distribution of the first two principal components (explaining 76% of variance within the soil spectral data).

We agree, however, that Extended Fig. 2c shows that the C fraction dataset should be expanded to cover more variability of the LUCAS VNIR dataset, potentially leading to a larger model applicability domain.

To address the reviewer's concern, we have included a statement on limitations and future research (also in line with comment #8), see L493 – L503.

7. The SOC fraction dataset lacks samples with high fine fraction and high MAOC content, which likely affects the modeling of MAOC saturation. Throughout their entire dataset, the MAOC content never exceeds 50 g kg⁻¹ (Extended Data Figs. 3 and 6). This limitation likely stems from the absence of high MAOC soils in their SOC fraction data, as their random forest model would not predict MAOC levels beyond the maximum observed in the training data. A recent study by Begill et al. (2023) has identified the presence of soils with high MAOC and high fine fraction in European agricultural systems Europe. Missing these soils could lead to an underestimation of the effective MAOC capacity, affecting the model's accuracy regardless of the modeling approach, and potentially skewing the estimation of risk factors. 2).

The reviewer has a good point here, identified by the Begill et al. work. However, we note that our local-regression method does not limit its predictions to the range of MAOC in the calibration set, as opposed to a random forest regression (not used in this study). That is, the local-regression method selects samples from the calibration set that have the closest resemblance to the target sample and then a weighted partial least squares regression is performed.

From our point of view, it is not necessarily true that high MAOC would have improved the results in terms of better representation of EU soils. In fact, high MAOC soils can potentially bias the analysis. I.e. they could leverage the entire regression (leading to overestimates for low MAOC values) if they are not properly weighted. This effect was one of the reasons we decided to propose a disaggregation approach based on pedo-climatic clusters, which aims to group for similarities avoiding the leverage effect.

In the dataset of Begill et al. (2023), agricultural soils high in MAOC are mostly characterized by Gleysols and Stagnosols. For a pan-EU assessment they are mostly of relevance in some Atlantic and Arctic biogeographic regions (Ibáñez et al. 2013), so we recognized that they might be important in regional calculations of MAOC estimates (L259-L262). We acknowledge that these soils exist and their absence in our data may lead to an underestimation in specific clusters, where those soils are present. Therefore, we have elaborated our discussion section with a statement on future research needs to extend the LUCAS sampling to those soils, see L393 – L395.

8. The small sample size may also contribute to the relatively low predictive power of the model in estimating MAOC (Extended Data Fig. 2; Line 127 to130). The data cloud distribution for the measured and predicted variables is very wide. The VNIR spectra do not appear to be good predictors of C fractions and SOC content. Given this sample size, it is challenging to assess whether the MAOC model is effective across different pedo-climate clusters.

We want to emphasize that the diagnostics of predictive performance should be interpreted in the context of the number of predicted values. That is, the scatterplot masks the density of points. One common method to deal with this visualization issue is the use of hexbin plots, where points are grouped in a hexbin polygon that is subsequently colored by the number of points that fall within that polygon (count), as shown in Figure 6 below.

Figure 5 – Reproduction of Extended Data Fig. 2b: Predicted POC + MAOC versus measured SOC for the sites selected after determination of the model applicability domain. The difference compared to the original figure lies in that points have been grouped into hexbins with their number indicated by the color scale.

From this figure, it is clear that the outer points of the cloud are less representative of the dataset than it appears from Extended. Data. Fig. 2. Additionally, in terms of the mean prediction metrics, the R^2 value is 0.59, with an RMSE of 8.8, a CCC of 0.76 (which is the closeness to the 1:1 line and can be interpreted on the same scale as R^2) and a low bias. We believe these values are within quite good for a dataset that spans a wide geographic extent, number of samples and range in SOC. We also refer to Supplementary Figure 1 where the prediction performance of C fractions and SOC are reported on an independent test set.

We agree that our initial description was too optimistic, particularly on POC estimates, which we have revised (L134-L140). We now also acknowledge that our dataset has limitations and refer to the lines we added based on comment #6 (L493 – L503).

9. I recommend that the authors address the potential limitations and implications associated with the size of their SOC fraction dataset. While their research undoubtedly represents the forefront

of the discipline, discussing these limitations will not only acknowledge the current dataset's potential constraints but also provide valuable insights for advancing future research in this area.

We thank the reviewer for recognizing the value of our analyses, and agree with the reviewer that acknowledging the limitations of our dataset used, and thus our inference, is important. We have therefore added sections that discuss this limitation (as also suggested by the editor) in the main text on L249-251, L259-L264, together with insights for advancing future research (L393-L397).

Reviewer #2 - Specific comments:

10. Beginning at line 130, but generally concerning the methodology for Δ SOC. Were the changes in SOC estimated or were they measured? My impression from reading the De Rosa et al. (2023) is that the Δ SOC were calculated using paired SOC measurements. However, Δ SOC was frequently referred to as estimated or predicted (e.g., line 130, 148). It would be helpful to specifically state how this parameter was calculated and clarify the methodology.

We thank the reviewer for pointing out that the reference to previous work in De Rosa et al. (2024) was insufficiently described. Indeed, the paper regressed the Δ SOC, derived from paired SOC measurements, against a set of covariates (clay, land use, SOC, precipitation and temperature, as well as coefficients of variation of precipitation and temperature on an annual basis). This regression was then used to estimate Δ SOC across the EU. We used these estimates in our analysis to evaluate their relationship against MAOC. In revision of the manuscript, we have now added text to clarify the methodology used to estimate Δ SOC based on De Rosa et al. (2024), to, see L559-L567.

11. Beginning line 193, it is evident that the Wsat provided a better fit for the predicted MAOC data compared to Hsat. This non-linear model also predicts lower effective MAOC capacity than those reported in past studies (Feng et al. 2013; Georgiou et al. 2022; line 216 to 221). These results could reflect the data pattern where the predicted MAOC remain relatively flat in soils with high fine fractions (Extended Data Figure 4). Could this pattern be associated with the lack of soils with high fine fractions and high MAOC? I agree with the authors that the LUCAS sampling design is sufficient to capture the most abundant soil types in European (line 222). However, there remains the concern that their SOC fraction samples might not be a good representation of European soils. If there is an artificial threshold due to the limited sample size, then is it fair to conclude that Wsat is an effective method for estimating effective MAOC capacity?

We agree with this point and we have included the upper limit discussion in the main text (L249-251, L259-L264). We also agree with the fact that Wsat provides a better fit compared to the Hsat method, although our aim was not to provide the best fit, but rather to compare methods and the concept behind (see reply to rev#1, comment #2 and Fig. 3 in main text). This has hopefully become more clear with changes to the discussion (L349-L367), based on comment 13. About the representativeness of data, please, we refer to our responses on comment 6-8.

12. Beginning at line 300, but for the entire SOC risk index paragraph. A table for the risk index categories & their combination of MAOC saturation and change in SOC may be beneficial for future readers.

Based on the reviewers' comment we have elaborated what was Suppl. Table S3 and placed it in the main text to support the discussion (L361-L367), which is now referred to as Table 1.

13. Line 294, I wonder why the median MAOC saturation was chosen as the classification criterion.

We thank the reviewer for raising this issue and refer to comment #15 that provides details on this decision.

14. Line 306, it might not be fair to refer to these soils as "MAOC saturated". They are more MAOC saturated than those in HE and LR, while they might not approach 100% saturation.

We agree and thank the reviewer for having pointed out this ambiguity! We have rephrased our interpretation of the risk index, based on comment #1 of rev1, where the renaming of the classes also lead that we have corrected this reference (L331).

15. Line 317, the similarity observed between three regression methods could also be explained by the use of median MAOC saturation as the classification criterion. In each cluster, while the regression methods influence the estimated degree of saturation in percentage, they do not affect the placement of a sample within the distribution of other data points. Thus, a soil will be categorized into either more saturated or less saturated group regardless of the regression methods.

We thank the reviewer for raising this issue. We agree with the fact that using the median MAOC saturation for each method imposed a limitation on comparing the relative effects of the methods on the risk index.

We have therefore re-calculated the risk index using the median MAOC saturation across the three regression methods (a MAOC saturation of 68%). Although the median degree of MAOC saturation (68%) is arbitrary, there is no scientific consensus yet on a generic threshold of MAOC saturation where C accrual diminishes and C losses become more likely.

We have therefore chosen for this approach since it fits with our aim to compare the effect of estimating MAOC saturation under different assumptions (Fig. 3 in main manuscript) and the implications this has on subsequent interpretations (risk index).

We have added an interpretation on these differences, using Table 1 (comment #12 above) and Extended Data Fig. 7. A convergence of evidence approach (i.e. the measure of agreement between regression methods) now informs interpretation (L349-L360) and future research directions (L395-L397).

16. I wondered whether the authors have plotted the Δ SOC against the estimated degree of saturation? According to the theory behind MAOC saturation, soils with lower MAOC saturation

would likely gain SOC more rapidly than more saturated soils (Georgiou et al. 2022). Similarly, more MAOC saturated soils might lose SOC more quickly. I think the authors have a rare opportunity to explore these hypotheses using a comprehensive regional dataset.

We thank the reviewer for this suggestion! We agree and have included a reference in the main text where this mechanism was discussed (L370-L376) together with a reference to the new Extended Data Figure 9.

Reviewer #3 (Remarks to the Author):

We thank reviewer #3 for reviewing the manuscript and improving it with suggestions.

References

- Begill, N., Don, A. and Poeplau, C., 2023. No detectable upper limit of mineral-associated organic carbon in temperate agricultural soils. *Global Change Biology*, 29(16), pp.4662-4669.
- Cotrufo, F.M., Lavellee, J.M., Six, Lugato, E. 2023. The robust concept of mineral-associated organic matter saturation: A letter to Begill et al., 2023. *Global Change Biology*, 29, pp.5986-5987.
- FAO; IIASA. Harmonized World Soil Database version 2.0. FAO; International Institute for Applied Systems Analysis (IIASA); Rome; Laxenburg, January 2023. ISBN 978-92-5-137499-3. doi: 10.4060/cc3823en. URL <http://www.fao.org/documents/card/en/c/cc3823en>
- Feng, W., Plante, A.F. & Six, J. Improving estimates of maximal organic carbon stabilization by fine soil particles. *Biogeochemistry*, 112, 81-93 (2013).
- Georgiou, K., Jackson, R.B., Vindušková, O., Abramoff, R.Z., Ahlström, A., Feng, W., Harden, J.W., Pellegrini, A.F., Polley, H.W., Soong, J.L. and Riley, W.J., 2022. Global stocks and capacity of mineral-associated soil organic carbon. *Nature Communications*, 13(1), p.3797.
- Ibáñez, J.J., Zinck, J.A. and Dazzi, C., 2013. Soil geography and diversity of the European biogeographical regions. *Geoderma*, 192, pp.142-153.
- Lugato, E., Lavellee, J.M., Haddix, M.L., Panagos, P. and Cotrufo, M.F., 2021. Different climate sensitivity of particulate and mineral-associated soil organic matter. *Nature Geoscience*, 14(5), pp.295-300.
- Viscarra Rossel, R.A., Webster, R., Zhang, M., Shen, Z., Dixon, K., Wang, Y.P. and Walden, L., 2024. How much organic carbon could the soil store? The carbon sequestration potential of Australian soil. *Global Change Biology*, 30(1), p.e17053.

Reviewer #1 (Remarks to the Author):

1. I have read the responses from the authors and the revision. I have the same concerns as I did before. The authors' responses were rather dismissive and the revisions were minimal.

We regret that the reviewer found our responses dismissive and considers them minimal. From our point of view, we have reviewed the comments of rev#1 rather thoroughly, supported by references to the literature in the discussion about the alternative regression method (see response 4) and data analysis (as recommended, we extracted the FAO soil types for our data, which showed larger variability of soil pH within soil type clusters compared to the pedo-climatic clusters that we have used. Thus, we found that using FAO soil types for clustering provided less spatially-refined information compared to the measured pH and remote sensing data which we used to formulate pedo-climatic clusters, see Annex I for details of our response in round 1). We regret that rev#1 found our comments dismissive, particularly since we included the suggested methodology in the supplementary material, and saw its value in providing an upper estimate (see response 4).

Additionally, we appreciated rev#1's remaining comments in round 1 and found they helped improving our manuscript. Based on them, we have re-written the sections of the risk index in the introduction, main text and methodology, aided by the new Extended Data Fig. 2 (L90-94, L298-L302, L332-L335, L580-L586). We also better acknowledged previous literature in the manuscript as suggested, L104-L106, L203-L205, L209-L211, as specified in the response of the first review round.

2. Briefly, in my view, the manuscript lacks scientific novelty and innovation for it to be published in Nature Comms. The idea that MAOC capacity is an emergent property from pedoclimatic conditions isn't new - as I noted previously. In their responses, the authors argue that they are adding their 'own contribution to the topic', but in my view the contribution to the scientific idea is minimal. Sure, they propose a 'risk index', but again, as I noted previously, the index is somewhat redundant and with little scientific value.

We would like to highlight that our work, for the first time, offers an analysis where the MAOC saturation concept is connected to temporal SOC losses at continental scales.

With regards to the pedo-climatic clustering: in our response, and in the manuscript, we acknowledged Georgiou et al. (2022), who divided soils in two classes on the basis of clay minerals activity, and Viscarra-Rossel et al. (2023) whose analysis relied on major soil types. Our approach differs in that it considers pedo-climatic conditions based on measured variables (pH), and those derived from earth observation (landform, net primary productivity, aridity). We have acknowledged this in the manuscript based on revisions in round 1 (L104-L106), now having added Georgiou et al. 2022.

Briefly, the scientific novelty and innovation of our manuscript lies in:

- i.) using visible-near infrared spectroscopy data from the LUCAS soil dataset to obtain the MAOC content for 6,548 samples, expanding from the legacy data available for Europe (see response 8 to rev#2 below),
- ii.) showing that no unique maximum MAOC capacity should be inferred from the fine fraction content only, but it should be considered an emergent ecosystem property from the pedo-climatic conditions
- iii.) providing new evidence on calculating C saturation through different approaches (see Fig. 3 in the main text)
- iv.) formulating a risk index that goes beyond the degree of MAOC saturation and assesses the status of the exposed SOC in agricultural lands.
- v.) we tested based on our extensive continental dataset whether soils high in MAOC saturation lead to more rapid losses, as shown in a global synthesis (Georgiou et al. 2022), see Extended Data Fig. 9 and Suppl. Table S3 in manuscript

3. The authors appear to insist that what they are proposing in terms of the pedoclimatic clustering is novel, but they do not provide solid arguments, and in fact, they tend to misplace or misinterpret citation to previous work.

Regarding the novelty of pedo-climatic clustering, it differs from previous approaches for the inclusion of climatic, ecosystem production and geomorphological characteristics. Therewith it provides additional evidence, within a European context with contrasting conditions to Australia (Viscarra-Rossel et al. 2023), that parameter estimates for the effective MAOC capacity are different between pedo-climatic clusters. This method should also be interpreted in context of the other study aims (see the bullet list in response 2).

As the reviewer did not provide any example where previous work is misplaced or misinterpreted, we are unable to respond/address this comment.

4. Regarding the fitting of the regressions and their comparisons. I maintain that the comparisons are rather insignificant because at least two of their functions are known to produce highly uncertain results. The authors argue against using the frontier method by misinterpreting the rationale for that method. For example, they write 'An assumption from the envelope method is that all high concentration MAOC points are accurately estimated...' - this is incorrect. The bootstrapped frontier method does not make that assumption and it provides confidence bands, just like the quantile method, but unlike the quantile method it does not produce underestimations.

The reviewer is misrepresenting here our revision. In fact we did not “argue against” but rather we agreed to include the frontier line method in our analyses since, given its characteristics, it can provide an upper estimate of the effective MAOC capacity. We opted to report the results from the frontier method in the supplementary material for the reasons highlighted above. We briefly lay out here the reasons for our decision to retain only the original three methods in the main text:

- the parametric nature of the quantile regression allows for comparisons with previous literature in which it has been used (e.g. Hassink et al. 1997; Six et al. 2002; Feng et al. 2013; Georgiou et al. 2022; Six et al. 2023) as opposed to the non-parametric frontier line analysis
- in line with previous studies, we preferred a more conservative estimate of the upper boundary based on the 90th quantile. Indeed this leads to a number of samples where the MAOC saturation exceeds 100% (due to underestimation), of which the magnitude depends on the three methods applied in the manuscript. For these cases we found it reasonable to set the MAOC saturation to 100% (as per the methodology) since they have been estimated as exceeding the soil's physical capacity.
- Our 'Wsat' method is similar in its theoretical basis, i.e. it assumes a non-linear texture dependent MAOC concentration (see Fig. 3 in main text)

Regarding the rev#1's statement: 'are known to produce highly uncertain results', we don't fully understand what rev#1 refers to. It would have been useful if rev#1 provided references/examples that supported this statement and allowed for an evidence-based discussion. Hassink's method has been used extensively in the previous literature (see references above). We align with that and provide a variation (Msat) as well as a novel non-linear model that better fits the upper boundary (Wsat). See Fig. 3 and methodology section.

Lastly, we don't see how the reasoning of rev#1 can hold up, since if the frontier line method does not produce underestimations, the mean regression line would not be lower than the upper boundary of the data. Indeed, the frontier line analysis is specifically aimed at estimating the maxima of the data as clearly highlighted in Viscarra-Rossel et al. (2023). Therefore, it assumes that the upper boundary, i.e. high concentration MAOC samples, don't have an associated error (since all general regression approaches have this assumption). The bootstrapping corrects slightly for this effect but the correction is marginal (see Fig 1). That is, the mean regression line lies in almost all cases on or above the maxima. We preferred a more conservative estimate of the upper boundary based on the 90th quantile, in line with previous studies.

Figure 1- Copy of Figure 4 in Viscarra-Rossel et al. 2023 that shows “frontier lines and their 95% confidence intervals fitted (a) using all 5089 observations, and (b) by soil type, represented by the orders of the Australian Soil Classification (Isbell, 2016). The frontier lines fitted and mineral-associated organic carbon (MAOC) stocks are displayed in their back-transformed original units.”

In summary, we appreciated the suggestion of the reviewer, acknowledging that any method to fit data is dependent on some assumptions, giving more or less weight to possible associated errors in the data (e.g. the choice of the quantile in our case). Therefore, we saw value in the proposed method as an upper estimate and this is why we included it in the supplementary material. We have slightly re-written the reference to this figure in the main text to clarify our rationale (L220 – L226).

5. I also have some concerns with the fractionation approach and the spectroscopy, but I will leave this here. If I were reviewing this for a soil science journal, I would be more critical about some of the methodologies used, including the clustering approach and the spectroscopy.

We regret that the comments of rev#1 are of anecdotal nature, making it difficult to have a constructive discussion and/or improve the manuscript. We respond briefly on the three topics:

The LUCAS C fraction data has been used in multiple publications (e.g. Cotrufo et al. 2019; Lugato et al. 2021; Georgiou et al. 2022; Hansen et al., 2023) and the fractionation was done based on a well-established published methodology (Cambardella et al. 1992). Indeed, recent studies found that the estimation of the POM fraction by size fractionation includes some OM associated with coarse and heavy (i.e. with sand-size) particles (Schweizer et al. 2021; Cotrufo et al. 2023; Leuthold et al. 2024). For this reason, we employed the Msat method to investigate the effect of accounting for this (see methodology section). The fractionation of MAOC has been shown to be reproducible between density and size fractionation (Leuthold et al. 2024).

Similarly, we don't see ground for the reviewer's comment on our soil spectroscopy methodology, as the regression method is based on publications and existing software packages (Ramirez-Lopez et al. 2013; Summerauer et al. 2021), similarly so for the outlier identification using the *F*-ratio method (Martens & Naes, 1989; Leifeld, 2006; Dangal et al. 2019). Furthermore, the sum of the estimated C fractions aligned well with the measured SOC for the LUCAS dataset that spans a wide geographic extent, has a large number of samples and range in SOC (Extended Data Fig. 3 of manuscript).

Regarding the clustering, the *k*-means algorithm (Hartig and Wong, 1979) is widely used in the literature and we applied it to measured variables (pH), and those derived from earth observation (landform, net primary productivity, aridity). All data sources were from published sources (see manuscript for details) and scaled to unit variance. A known issue of *k*-means is its dependence on the initial allocation of cluster centroids, which we corrected by running the clustering procedure 100 times, under different random seeds. Within each procedure the initial centers were allocated another 100 times using the 'nstart' argument in the *kmeans()* function of the 'cluster' R package, allowing for relatively robust estimates. We refer to the methodology section for further details.

6. More generally, however, my recommendation is that the paper lacks sufficient novelty and global scientific significance for publication in Nature Comms."

This comment of rev#1 took us rather by surprise, since there was no mention of a lack of global significance in round 1, and the reviewer seemed to find it hard to identify a unique element of novelty, but not to question it all together. In fact, rev#1 wrote: "*I found your paper a little bit 'too complicated' what is it about? the Msat and Wsat? is the spectroscopy a novelty too? or is it the derivation of the pedoclimatic regions? - the title seems to suggest that it is the risk index, but there is little on that. Perhaps you propose that it is about all of those - I would suggest that you reduce methodological complexity and focus your study on the most important aspect*".

We argue that the bullet list provided under response 2 constitute the scientific novelty of our study which we copied here:

Briefly, the scientific novelty and innovation of our manuscript lies in:

- i.) using visible-near infrared spectroscopy data from the LUCAS soil dataset to obtain the MAOC content for 6,548 samples, expanding from the legacy data available for Europe (see response 8 to rev#2 below),
- ii.) showing that no unique maximum MAOC capacity should be inferred from the fine fraction content only, but it should be considered an emergent ecosystem property from the pedo-climatic conditions
- iii.) providing new evidence on calculating C saturation through different approaches (see Fig. 3 in the main text)
- iv.) formulating a risk index that goes beyond the degree of MAOC saturation and assesses the status of the exposed SOC in agricultural lands.
- v.) we tested based on our extensive continental dataset whether soils high in MAOC saturation lead to more rapid losses, as shown in a global synthesis (Georgiou et al. 2022), see Extended Data Fig. 9 and Suppl. Table S3 in manuscript

Overall, our paper adds to the discussion on how to calculate and interpret MAOC saturation as an indicator of SOC status. These findings are also relevant within an EU policy context aimed to increase the land C sink, as outlined in the introduction. The approach and our results are however informative for soil C management in general, whether in a scientific, private sector or policy context.

Reviewer #2 (Remarks to the Author):

7. I appreciate the authors' response and revision in response to my previous comments. They included new text on the potential limitation of their methods and the future research directions. Their clarification about some methods and the recalculation of the risk index are also welcome. I have some follow-up regarding the authors' responses to my comments and the next text:

Regarding comment 6 from the previous review, the authors' responses make sense, and their arguments are well laid out in the methods section (L493-L503). Personally, the sample size of the fractioned data is still small ($n = 240$) for a large-scale study covering Europe. Compared to my other comments (6, 8, 9, and 11), my main concern was about the sample size and potential consequences. Thus, I would suggest that the authors acknowledge it in the main text.

We thank the reviewer for the comment and agree that an acknowledgement in the main text is appropriate. We have now added this on L 247 – L 249.

8. Regarding the comment 7, I do not agree with the statement that high MOAC soils in Begill et al. (2023) are mostly Gleysols and Stagnosols. My group re-analyzed their data, which are publicly available, and yes, their dataset does include Gleysols and Stagnosols, indicative of hydric

conditions that contribute to OM accumulation that are distinct from the hypothesis of MAOC accumulation used in the POC/MAOC framework. However, high MAOC content soils represented in that dataset are not exclusive to the Gleysols and Stagnosols. Six et al.'s reanalysis also supports this assessment (DOI: 10.5194/soil-10-275-2024). Furthermore, the global dataset from Georgiou et al. 2022 contained many data points with MAOC above 50 g C kg⁻¹. Thus, there is still the concern that the maximum MAOC content in LUCAS (Line 250) might be an artifact due to sampling design and sample size. In sum, the issue is bigger than the soils with oxygen limitations (as portrayed by the authors L260-264).

We appreciate the reviewer's comment and acknowledge that we made an oversimplified statement. We have made changes to the text to avoid misinterpretation. The arguments of gleysols/stagnosols (Six et al. 2023) and exceedance of 50 g C kg⁻¹ (Begill et al. 2023) are now separated better. We also added a reference to our new findings in the supplementary material L251-L267.

The new findings are the result of our investigation to fully address the comment of the reviewer: we have investigated the effect of including all relevant legacy data from Georgiou et al. (2022) and Begill et al. (2023):

We selected all points from Georgiou et al. (2022) according to two criteria: 1.) located within the EU but not part of LUCAS, 2.) were under cropland or grassland land use (n = 146).

The Begill et al. (2023) data are available with protected coordinates (i.e. coordinates have been 'randomly' sampled within a radius of 4 km from the sampling location), (n = 113).

Figure 1 shows the boxplots of the MAOC compared to the data used in the manuscript (LUCAS VNIR, n = 6,548).

Figure 2 - Comparison of the mineral-associated organic carbon (MAOC) data based on LUCAS VNIR data used in our study (n = 6,548) and the datasets reported in Georgiou et al. (2022) (n = 149) and Begill et al. (2023) (n = 113).

We extracted the pedo-climatic cluster association of these points based on the raster in what is now Suppl. Fig. 11 and then investigated the effect of including the legacy data by:

- Reproducing Extended Data Figures 4-6
- Reproducing Figure 4 of the main text

Figure 4 in the main text is based on the parameter estimates from Extended Data Figs 4-6 and serves as input to computing the index.

Here we list a reproduction of Extended Data Figs. 4-6 of the main manuscript with the legacy data included.

Figure 3 | Reproduction of Extended Data Fig. 6 in the manuscript: non-linear 90th quantile regression to determine the effective MAOC capacity for the whole saturation method (Wsat), intercept was restricted to 0. For ease of cross-comparison with alternative methods, parameter estimates have been converted to g MAOC in kg⁻¹ of fine fraction. Datasets correspond to Georgiou et al. (2022), n = 146, Begill et al. (2023), n = 113, and the LUCAS MAOC data based on VNIR spectra (n = 6,548).

Figure 4 | Reproduction of Extended Data Fig. 5 in the manuscript: piece-wise linear 90th quantile regression to determine the effective MAOC capacity for the mineral saturation method (Msat), slope of the second linear equation (after the breakpoint, indicated by the vertical dashed line) was restricted to 0. For ease of cross-comparison with alternative methods, parameter estimates have been converted to g MAOC in kg^{-1} of fine fraction. Datasets correspond to Georgiou et al. (2022), $n = 146$, Begill et al. (2023), $n = 113$, and the LUCAS MAOC data based on VNIR spectra ($n = 6,548$).

Figure 5 | Reproduction of Extended Data Fig. 4 in the manuscript: linear 90th quantile regression to determine the effective MAOC capacity for the Hassink method (Hsat) by each pedo-climatic cluster, the intercept was restricted to 0. For ease of cross-comparison with alternative methods, parameter estimates have been converted to g MAOC in kg⁻¹ of fine fraction. Datasets correspond to Georgiou et al. (2022), n = 146, Begill et al. (2023), n = 113, and the LUCAS MAOC data based on VNIR spectra (n = 6,548).

Our short interpretation is as follows:

- The data from Georgiou et al. (2022) lay within the limits of the LUCAS VNIR data (Fig. 2 and Fig. 3-5). The Begill et al. (2023) data lay partially within limits of the LUCAS VNIR data when: i.) plotted as a function of the fine fraction (clay + silt), see Figs. 3-5, and ii.) when MAOC was expressed as MAOC content in the fine fraction (lower left panel, Fig. 2).
- The inclusion of the legacy data affects parameter estimates for a minimal set of clusters only (mostly cluster 8 and 11) (Figures 3-5), illustrating that separating the data by pedo-climatic clusters minimizes the leverage compared to pooling all the data. That is, a large

number of clusters remained unaffected by including the legacy data. For example cluster 4, located in the Iberian peninsula.

Additionally, reproducing Fig. 4 of the main text showed that the differences in parameter estimates don't translate through in large changes for the MAOC saturation to fine fraction relationship (see Fig. 6 in this response), and consequently the SOC risk index.

Note that there are also slight differences for clusters that did not have any associated legacy data (e.g. cluster 4). This is because we found a small mistake in the code, where a selection of samples was excluded in the calculation of Fig. 4 (of the main text). We have now corrected for this throughout the manuscript. It did not affect the overall findings.

In the end, we decided not to include these legacy data in our analysis, due to the different sampling years, sampling depths and analytical methods used in the data of Georgiou et al. (2022) and Begill et al. (2023). We have added this analysis to the Supplementary material (Suppl. Figs 4-8) together with a reference in the main text (L265-L267).

Figure 6 | Degree of mineral-associated organic carbon (MAOC) saturation (MAOC / effective MAOC capacity x 100%) as a function of fine fraction (clay + silt, %). Values below 100% indicate a saturation deficit relative to the cluster-dependent effective MAOC capacity. Fine fraction content has been binned by intervals of 10%, points and lines represent the mean degree of MAOC saturation \pm standard deviation for each bin. The y-axis is on a \log_{10} scale. Datasets correspond to LUCAS MAOC data based on VNIR spectra used in our study ($n = 6,548$) and the LUCAS VNIR + legacy data (Georgiou et al. 2022; Begill et al. 2023).

9. I appreciate the renaming of the four categories for the risk index, as they are more intuitive than the last version. I still have some trouble differentiating High Risk from High Hazard. In fact, it is hard to remember which is which. In the main text, could the authors elaborate on where the names came from?

We thank the reviewer for the comment. Both index classes (high risk and high hazard) correspond to raster cells that have negative Δ SOC. High hazard is when that raster cell shows lower vulnerability (i.e. MAOC saturation below median). In case of high risk, the raster cells are above the median MAOC saturation, having a high risk of larger SOC losses. We have now added additional text to introduce the risk index categories also with reference to Extended Data Fig. 2. See L332 – L335.

10. Regarding the table about SOC risk index, moving it from the supplement to the main text is a good starting point but does not address the full extent of the original comment. The main idea behind the inclusion of this table is to make this concept as intuitive and understandable as possible for future readers. There are a few issues with the table: 1) there is enough space to expand the abbreviation of column two to include the full name of the risk category; 2) one of the abbreviations should be “HN” rather than “LH”; 3) Perhaps change the title of the third column from Mha to Area (Mha); 4) It may be helpful to show the combination of change in SOC and MAOC saturation that corresponds to each category. For example the positive change, no change, and negative change in SOC described in lines 325-326, as well as above and below MAOC saturation in lines 323-325

We thank the reviewer for thinking along on how to improve the manuscript! We have implemented the suggested changes to the table. We have also changed i.) the order of the rows for better interpretation and ii.) the color scheme in of the risk index quadrat in Fig. 5, where red and yellow now correspond to negative Δ SOC values and green and blue for positive Δ SOC. We thought these colors to be more intuitive for interpretation.

Reviewer #2 (Remarks on code availability):

11. The codes were mostly for a published paper. I cannot find codes specific to this study.

We appreciate the call for open research. Indeed, that repository hosts publications from the same dataset on C fractions. The agreement with the senior editor dr. Buongiorno was to host the code on that repository once the manuscript is accepted for publication.

Reviewer #3 (Remarks to the Author):

We thank the rev#3 for taking the time to review the manuscript.

Annex I – response to comment 3 of reviewer 1 in first round of revisions

3. Regarding the derivation of the pedoclimatic zones, this makes sense, of course but I wonder if the authors tried to more simply use soil type? Soil types represent pedoclimatic regions - as Jenny's *cl,o,r,p,t* model shows. This might reduce complexity and be more intuitive and useful.

We agree with reviewer 1 that reducing complexity and the use of intuitive soil classes would be useful. Therefore, we have looked at the availability of EU soil type data based on the World Reference Base (WRB) soils from the FAO (FAO, 2023). The map available appeared relatively coarse, potentially missing spatially refined information on specific soil/soil-forming conditions. This is illustrated by the large spread of the aridity, landform, NPP and pH properties within the WRB classes. For example, the more common soil type classes (CMdy, CMeu) have a range of 4-8 for pH in H₂O (on a log-scale).

Figure 3 - World reference base (WRB) soil classes (FAO, 2022). a.) spatial distribution for the LUCAS soil data that have VNIR measurements. b.) boxplots of the cluster variables by WRB class.

We point out that one of main intentions of this work is to test whether effective MAOC capacity is an emergent ecosystem property of pedo-climatic conditions. Therefore, we used measured variables (pH), and those derived from earth observation (landform, NPP, aridity). This method of spatial classification could be more reliable than solely extracting information from a relatively coarse soil type map, which could introduce higher uncertainty due to spatial interpolation.

To emphasize its rationale, coherence and logic, we have re-written the section on clustering. Specifically, we have moved the methodological details on transferring the clusters on the EU raster (L298—L300) and refer to the Suppl. Figures (5 and 6) for the interested reader.

References

- Begill, N., Don, A. and Poeplau, C. No detectable upper limit of mineral-associated organic carbon in temperate agricultural soils. *Global Change Biology*, **29**, 4662-4669 (2023).
- Cambardella, C.A. & Elliott, E.T. Particulate soil organic-matter changes across a grassland cultivation sequence. *Soil Sci. Soc. Am. J.* **56**, 777-783 (1992).
- Feng, W., Plante, A.F. & Six, J. Improving estimates of maximal organic carbon stabilization by fine soil particles. *Biogeochemistry* **112**, 81-93 (2013).
- Cotrufo, M.F., Ranalli, M.G., Haddix, M.L., Six, J. & Lugato, E. Soil carbon storage informed by particulate and mineral-associated organic matter. *Nat. Geosci.* **12**, 989-994 (2019).
- Dangal, S.R.S., Sanderman, J., Wills, S. & Ramirez-Lopez, L. Accurate and precise prediction of soil properties from a large mid-infrared spectral library. *Soil Syst.* **3**, 11 (2019).
- Georgiou, K., Jackson, R.B., Vindušková, O., Abramoff, R.Z., Ahlström, A., Feng, W., Harden, J.W., Pellegrini, A.F., Polley, H.W., Soong, J.L. & Riley, W.J. Global stocks and capacity of mineral-associated soil organic carbon. *Nat. Commun.* **13**, p.3797 (2022).
- Hansen, P.M. Even, R., King, A.E., Lavalley, J., Schipanski, M. & Cotrufo, M.F. Distinct, direct and climate-mediated environmental controls on global particulate and mineral-associated organic carbon storage. *Global Change Biology*, **30**, e17080 (2023).
- Hassink, J. The capacity of soils to preserve organic C and N by their association with clay and silt particles. *Plant Soil* **191**, 77-87 (1997).
- Leifeld, J. Application of diffuse reflectance FTIR spectroscopy and partial least-squares regression to predict NMR properties of soil organic matter. *Eur. J. Soil Sci.* **57**, 846-857 (2006).
- Lugato, E., Lavalley, J.M., Haddix, M.L., Panagos, P. & Cotrufo, M.F. Different climate sensitivity of particulate and mineral-associated soil organic matter. *Nat. Geosci.* **14**, 295-300 (2021).
- Martens, H. & Naes, T. *Multivariate calibration*. John Wiley & Sons (1992).
- Ramirez-Lopez, L., Behrens, T., Schmidt, K., Rossel, R.V., Demattê, J.A.M. & Scholten, T. Distance and similarity-search metrics for use with soil vis-NIR spectra. *Geoderma* **199**, 43-53 (2013).
- Ramirez-Lopez, L., Stevens, A., Viscarra Rossel, R., Lobsey, C., Wadoux, A. and Breure, T. (2022). resemble: Regression and similarity evaluation for memory-based learning in spectral

chemometrics. R package Vignette R package version 2.2.1. Accessed at 19-12-2023: <https://CRAN.R-project.org/package=resemble>.

Shenk, J., Westerhaus, M., & Berzaghi, P. Investigation of a LOCAL calibration procedure for near infrared instruments. *J. Near Infrared Spectr.* **5**, 223-232 (1997).

Six J., Conant R.T., Paul E.A. & Paustian K. Stabilization mechanisms of soil organic matter: implications for C-saturation of soils. *Plant Soil* **241**, 155–176 (2002).

Six, J., Doetterl, S., Laub, M., Müller, C.R. & Van de Broek, M. The six rights of how and when to test for soil C saturation. *EGUsphere* **2023**, 1-8 (2023). Smith, P. Managing the global land resource. *Proc. Royal Soc. B: Biol. Sci.* **285**, 202172798 (2018).

Summerauer, L., Baumann, P., Ramirez-Lopez, L., Barthel, M., Bauters, M., Bukombe, B., Reichenbach, M., Boeckx, P., Kearsley, E., Van Oost, K. & Vanlauwe, B. The central African soil spectral library: a new soil infrared repository and a geographical prediction analysis. *Soil* **7**, 693-715 (2021).

Viscarra Rossel, R.A., Webster, R., Zhang, M., Shen, Z., Dixon, K., Wang, Y.P. & Walden, L. How much organic carbon could the soil store? The carbon sequestration potential of Australian soil. *Global Change Biology* **30**, p.e17053 (2024).

“Revisiting the soil carbon saturation concept to inform a risk index in European agricultural soils”

Reviewer response – round 3

Reviewer #1 (Remarks to the Author):

Comment 1

Thanks to the authors for considering my comments and addressing them. Most of comments have been satisfactorily resolved. However, I still have concerns about Table 1. The High Hazard and No Risk groups appear to be mislabeled. According to L332-333, High Hazard is associated with SOC losses and MAOC saturation below median. In contrast, Table 1 shows that the High Hazard group is characterized by SOC gains. Additionally, some of the area estimates seem inconsistent. The text (L345-346) states that High Risk group covers an area of 43-83 Mha; however, these numbers do not match with those from the table ($27.5+2.5 = 30$ Mha). Given the importance of the risk index in the authors' novelty arguments, caution should be exercised to ensure the accuracy of this table and associated results, including Figure 5 and related supplementary figures.

We thank the reviewer for these sharp observations and point out these inconsistencies! Indeed, there was a mistake in the sense that the total area for each class did not add up to the correct number and we mislabelled the classes in Table 1.

After reviewing the table, we also found that the caption was insufficiently descriptive. The agreement/disagreement was based on the overlap between all three different methods. Therefore, the sum of the agreement/disagreement area is equal to the total area covered by the SOC index raster (187 Mha).

The percentages by SOC index class are calculated for one reference method (previously Msat, PBL) whereas the areas for each SOC index class in the text (43-83 Mha) refer to the ranges between the methods.

We have now corrected Table 1 and changed the reference method to NBL (since this is what we recommend to be developed in future work, L327-L335, see comment #7). The conclusions we draw from the table (larger agreement for HR and NH classes) still hold and the total area for each class (Mha) now correspond to the mean values provided in Suppl. Table 3.

Comment 2

Reviewer #1 (Remarks on code availability):

According to the authors, codes will be made available after acceptance. I could not review their codes at this stage.

We have now made the code available under this link, and can be reviewed. The repository is restricted for the duration of the review process.

Comment 3

Reviewer #2 (Remarks to the Author):

"I co-reviewed this manuscript with one of the reviewers who provided the listed reports. This is part of the Nature Communications initiative to facilitate training in peer review and to provide appropriate recognition for Early Career Researchers who co-review manuscripts."

We thank the reviewer for their comments to improve our manuscript.

Reviewer #3 (Remarks to the Author):

Comment 4

The authors propose a risk index as a combination of vulnerability (MAOC saturation) and hazard (Δ SOC) for agricultural soils. The novelty lies mainly in the methodology of estimating MAOC saturation. The authors' idea is based on the effective "biophysically achievable" MAOC capacity determined from clustering pedo-climatic conditions across Europe and using three different regression methods to estimate saturation separately for each cluster.

The problem of defining realistic carbon sequestration goals and focusing efforts on most promising areas is globally relevant, and the discussion of different methodologies to address this problem contributes to both science and policy development. From a policy perspective, the resulting index mapped over the EU could be considered the main contribution. From the science perspective the paper offers two main opportunities: 1) discussion of MAOC capacity as an emergent ecosystem property; 2) discussion of regression methods used to estimate capacity from observations. In my opinion, both topics are not fully addressed in the current manuscript and can be improved upon as detailed below.

1) The authors argue that the effective MAOC capacity cannot be determined solely based on the clay content. Previous studies shared the same argument and added clay mineralogy (Georgiou et al., 2022) and soil types (Viscarra-Rossel et al., 2023) to MAOC capacity estimation. Here the authors chose to stratify soils by pedo-climatic clusters based on pH, landform class, NPP, and aridity index. I would like to see more discussion of this choice of factors in terms of: a) Justification of factor selection. Adding or removing a factor can significantly change clustering results, therefore the readers should be able to see the logic of e.g. including landforms, or e.g. not including nutrients, total SOC or plant functional types.

We agree with the reviewer that the decision on which factors to include for the clustering is an important methodological component.

The rationale for including these factors was as follows: pH as a proxy of clay mineralogy and affecting SOC turnover (microbial composition); landform since the erosion/depositional setting affects preferential displacement of SOC fractions; NPP as a driver of saturation and aridity as a synthetic climate parameter.

We did not include plant functional types, given that our analysis concerns agricultural soils. Additionally, we did not use SOC and clay, since they are variables in the regression to estimate the effective MAOC capacity for each cluster. We avoided the use of soil-nutrients (in particular N) as stratification factor given the high correlation with SOC. We have added the rationale for this in the main text (L118-L121).

Overall, our number of clusters resulted in a similar number as other studies that aimed at estimating soil organic carbon content targets across the EU (Pacini et al. 2023), increasing confidence in the approximation.

However, based on the reviewers' suggestion for adding/removing cluster covariates, we have implemented a sensitivity analysis, we refer to comment 5 below.

Comment 5

b) Even more importantly: understanding the mechanisms behind the clusters. It is implied that the clay-MAOC relationship is modified by the pedoclimatic conditions, represented by the 4 abovementioned factors. But there is no discussion (or better quantitative analysis) of why that would be the case (and why the selected factors are deemed more important than the omitted ones).

c) While effects of the 4 pedo-climatic factors could be initially hypothesized, eventually the authors could analyze their results to see the individual effects of each clustering factor on the estimated effective MAOC capacity. Does low pH tend to increase or decrease it? Why? How are landforms affecting MAOC capacity? Could they be a proxy of mineralogy, or water redistribution, or erosion? Such analysis and discussion may greatly improve the scientific insight that the reader could draw from the paper, compared to a probably correct, but rather obvious statement that "clay alone is not enough".

We thank the reviewer for the view and interesting discussion.

However, our analysis treats landform, pH, NPP and aridity as controlling factors rather than independent variables. We can therefore not quantitatively express to what extent pH or landform affects the MAOC capacity.

Triggered by the reviewers' questions and comments, we have done a sensitivity analysis for the *k*-means clustering procedure as implemented in the manuscript. In particular, we have iteratively removed each of the four covariates used in the clustering, estimated the effective MAOC capacity and assessed the distribution of the effective MAOC capacity across clusters. For the purpose of the sensitivity analysis we have estimated the effective MAOC capacity by the NBL method (previously Wsat) only.

Our hypothesis was that the spread in β parameters, indicating the effective MAOC capacity, should be reduced when excluding a cluster covariate that helps to distinguish between the theoretical (based on fine fraction only) and the effective MAOC capacity (considered as an ecosystem property, accounted for by controlling factors). That is, if the cluster covariates did

not have an effect on determining effective MAOC capacity, their exclusion would not narrow the distribution in β parameters. In case they do not have an effect, the distribution across the clusters would be quite very similar in the leave-one-covariate-out experiment compared to the original.

Figure 1 Distribution of β parameters across all clusters from the leave-one-covariate-out sensitivity analysis. 'Original' is as reported in the main manuscript. The other boxplots show the distribution when that covariate was excluded from the clustering, where k indicates the total number of clusters. Model fitting was done as per the NBL method (previously Wsat), specified in the main manuscript. The center line of the boxplots is the median, the boxplot lower- and upper limits are equal to the first and third quartiles, respectively. The upper whisker extends 1.5 times the inter-quartile range from the upper limit, vice versa for the lower whisker.

Our findings showed that this was not the case (Fig. 1): the inter-quartile range of the β parameter distribution across all clusters reduces when removing factors from the k -means clustering. This indicates that these covariates help to distinguish between the theoretical (based on fine fraction only) and the effective MAOC capacity.

There are two main effects of covariates on reducing the theoretical maximum to effective maximum MAOC capacity: i.) aridity and NPP have a larger effect on the β parameter distribution, ii.) pH and landform also showed an effect although it was smaller. Whereas pH and landform tend to reduce the median effective MAOC capacity (i.e. their exclusion increases the median), aridity and NPP increase the median effective MAOC capacity.

Finally, the results show that our original approach led to a wider differentiation of the effective MAOM capacity across the EU agricultural soils. In line with the reviewer's suggestion, we have included a short discussion on the findings from the sensitivity analysis to discuss the effect of controlling factors on the effective MAOC capacity (see L213 – L217).

Comment 6

d) The authors' definition of effective MAOC capacity differs from Georgiou et al. (2022) and Viscarra-Rossel et al. (2023) not only in the selection of factors, but in a more fundamental way. While previous definitions focused on practically unchanging properties, i.e. texture, mineralogy and soil types, in this paper the estimation is based on dynamic properties i.e. NPP, pH and

aridity. Thus, in the previous studies, “MAOC capacity” was an invariant physical soil property, to which actual MAOC content could be compared, and a “stationary target” which land users may strive to achieve through improving their practices. In this paper, the authors use the same “MAOC capacity” term for an ecosystem property that is altered by climate change (aridity index, NPP), and land use (NPP and pH). I understand that authors used 20-year averages for aridity and NPP counting it as long-term climate. But let’s imagine we use their MAOC capacity as a target for a long-term land management project today. The result of the project may be evaluated in 20 years - a reasonable time for changes in slow-turnover carbon pools such as MAOM, and quite a significant time considering rapid climate change and even faster land management changes. By then we will have different aridity, different NPP, and possibly different pH, especially if there was a substantial change in land use practices. So, the same land area may get into a different cluster in 20 years, and its estimated MAOM capacity will change. We get a “moving target” instead of a stationary one. The situation may be particularly confusing in case our land management intervention is designed to increase SOC by enhancing C input to soils via larger NPP, e.g. by implementing cover crops and irrigation. In this case we will be by definition changing the MAOC capacity, rather than trying to reach it! I see the benefit of considering the whole ecosystem in realistic target-setting, but I’d like more clarity in definitions here. E.g. we should avoid the confusion of using the same terms “MAOC capacity” for a time-invariant soil physical characteristic and simultaneously for a time-dependent ecosystem property. The authors use the term “effective MAOC capacity”, which I think should be more clearly defined, especially in the context of the methodology potentially being used for agricultural land risk-assessment or climate target setting.

We thank the reviewer for this elaborate comment. We agree that the definitions of the (effective) MAOC capacity were not properly defined in the manuscript and have changed this accordingly throughout, in particular the introduction (see L59—L63, L64, L70, L74, L80).

However, from our point of view, the time-dependence is also partially valid for the studies in Georgiou et al. (2022) and Viscarra-Rossel et al. (2023). In a scenario where conditions would change substantially (e.g. a 1.5°C global warming scenario) , they will likely find another MAOC saturation threshold as observations may approximate another “theoretical maximum” and there is still incomplete knowledge to what extent the ‘theoretical’ and ‘effective’ MAOC capacity coincide. From our point of view, the discussion to what extent MAOC capacity is variable is still an open debate and our paper seeks to contribute to this discussion. For example, the points observed may not be truly saturated for limiting conditions (NPP, nutrient, microbial composition etc.) or sampling may not cover all the covariate space.

To bring it back to our analysis, we find a maximum effective MAOC capacity within clusters which is an emergent property determined by those conditions. In a scenario where the environmental conditions will change substantially within 20 years, we could re-cluster Europe but the relation MAOC saturation vs clay calculated in each cluster is still valid (e.g. MAOC saturation vs clay in cluster 7 will be applied to the new spatial extent of the cluster 7). This holds

true under the assumption that our methodology captures fundamental drivers of SOC dynamics (which also holds for Georgiou et al. 2022 and Viscarra-Rossel et al. 2023). From our point of view, our methodology is therefore scalable in time and space. Although it could be improved in case additional data becomes available and additional drivers/higher density of points, see L426—430.

Comment 7

2) I appreciate the authors' effort to investigate three different regression methods of estimating the effective MAOC capacity. However, it is rather unclear to me what the conclusion regarding these methods is. If anybody would like to replicate the proposed methodology for a different geographic region, would they need to re-examine all three methods? Or is there some insight on a preferable regression method that the authors would like to share based on their investigation?

We thank the reviewer for the comment. That goes hand in hand with the concept of saturation that we would like to debate. Is it only a single layer clay coating or organo-organobounds, often found in northern European sandy soils (that appears to oversaturate with boundary line approaches)? Our opinion is that, moving from a more theoretical to a more practical/pragmatic point of view, the NBL (previously W_{sat}) method can represent a better target for policy implementation and merits further research. We have now added this to the discussion at L327-335. We have also aligned Table 1 with the NBL method (see comment #1) for consistency with this recommendation.

Comment 8

3) Additionally, I would like to note that the whole calculation of MAOC saturation relies on the initial step of predicting MAOC from a limited number of measurements using VNIR spectroscopy data. I assume such prediction had a significant uncertainty and I wonder if that uncertainty was taken into account in the subsequent steps of MAOC capacity estimation and risk assessment. If it was not taken into account, then this should be done and presented in a quantitative way, as well as discussed.

The topic of a limited set of measurements has also been discussed in a previous review round based on a comment of rev#1 and rev#2. Based on that we added the following lines to the manuscript, where we state the limitations of the LUCAS dataset and the calibration dataset (L259-L265, L275-L276). Accounting for these limitations, we have determined the model applicability domain, based on the F -ratio method (Extended data Figure 3). This Figure shows how the C fractions predictions have been restricted based on the confidence in our predictions from VNIR spectra: they lie within the range of the calibration set in terms of the first two principal components of the VNIR spectra.

The fact that the predictions are restricted to the range of the calibration dataset is also shown in Fig. 2 below, where the particulate organic carbon (POC) over total SOC and the mineral associated organic carbon (MAOC) over total SOC against the C:N ratio have been plotted. Based

on theory, we would expect an increase in the POC:SOC ratio for higher SOC samples, since the MAOC fraction becomes saturated (Cotrufo et al. 2019; Georgiou et al. 2022). Higher MAOC / SOC on the other hand, is generally associated with higher C:N content (Cotrufo and Lavelle, 2022). Here it shows that the C fractions predictions ('P(F) < 0.99', black point) follow the expected trend and are in line with the calibration set ('C fraction', red points).

Figure 2 Diagnostic plots of the predicted mineral-associated organic carbon samples ($P(F) < 0.99$) and the comparison of the calibration dataset (C fraction-red points). Panel (a) shows the particulate organic carbon (POC) over the total soil organic carbon (SOC) plotted against the range of SOC. Panel (b) plots out the mineral-associated organic carbon (MAOC) over SOC versus SOC over the total soil nitrogen content (C:N ratio).

However, we do acknowledge that there is an associated uncertainty with the predictions regardless. To assess the effect of marginal uncertainties in our predictions, we have approximated the expected error based on the predicted POC+MAOC vs. measured SOC (Extended Data Fig. 3). Given the negatively skewed distribution of SOC, we calculated the mean absolute log error (MALE). The MALE is robust to outliers (high SOC values). MALE reduces the effect of large differences between the predicted and measured values and provides a better measure of the relative difference. That is, the exponential of the MALE (EMALE) represents the relative multiplicative error (once we subtracted 1). We assumed the error to be normally distributed around the mean prediction and that POC and MAOC contribute equally, so we divided the EMALE by two.

We then performed 500 simulations where we resampled the mean MAOC prediction (the MAOC values presented in the main manuscript) with a standard deviation (σ) represented by $(EMALE-1) \times MAOC$. See Fig. 3a for the distribution of the standard deviations across the MAOC range. Fig. 3b shows a detailed example of the simulated distribution of MAOC values for a case where the mean MAOC is equal to 15.6 g kg^{-1} . The kernel density plots of the 500 MAOC

simulations showed how for each simulation, the overall MAOC distribution approximated the density distribution of the MAOC presented in the manuscript (Fig. 3c).

Figure 3 a.) Distribution of the standard deviation (σ) used in the simulation of prediction errors, b.) detailed example of MAOC simulations for a single MAOC value, with the mean (μ) and standard deviation (σ) provided in text, c.) density distributions of the mineral-associated organic carbon (MAOC) simulations (black lines) and the original distribution (red line). Dashed lines indicate the 1st quartile, median and 3rd quartile, respectively of the original distribution

Based on these simulations, we followed the same methodology as per the manuscript.

Figure 4 This Figure is analogue to Fig. 4 in the main manuscript. However, it reports the overall distribution of 500 MAOC realizations (500 x n points in each fine fraction bin). Red circles represent the mean MAOC saturation values by each bin as reported in the main manuscript. We have restricted the y-axis to 100% here, since any differences above 100% would not affect subsequent inferences from the data. Note that the y-axis in the original figure is on the \log_{10} scale.

The results showed the following effect of simulated prediction uncertainty on the mean MAOC saturation:

- the mean estimates as reported in the manuscript (red circles) were in the majority of cases close to the median of the simulations. This indicates that the number of simulations was sufficient to approximate the variation due to the standard deviation

around the mean, since the joint distribution of the samples within each bin followed the normal distribution.

- The effect of simulated prediction uncertainty on the mean MAOC saturation appears partially to be smoothed out by averaging across fine fraction bins within a cluster. The inter-quartile range relative to median (or the original values, red circles) is small across the fine fraction range in most clusters. Although we have noticed two exceptions to this:

1. Some clusters, in particular with high MAOC content (clusters 1 and 15), show larger variability in MAOC saturation across the simulations. This is likely due to the multiplicative error context, where higher values have a higher associated error. However, there were differences between the estimation methods (BL, NBL, PBL). For example, PBL in cluster 13 also showed large variation between the simulations. In general, the PBL method showed larger variability, likely due to the determination of the 'breaking point' in the piece-wise linear quantile regression that adds additional uncertainty to parameter estimates.

2. As expected, fine fraction bins for which we have fewer data points, are affected in particular. These tend to be for low fine fraction content samples (bins 1-3).

For each of the 500 simulations, we then also calculated the MAOC saturation across Europe and the SOC index. We calculated the 5th and 95th quantiles of the MAOC saturation rasters (Suppl. Fig. 15) and the area for each SOC index class (Suppl. Table 3).

We have added both of these to the supplementary material with a reference in the main text (L376-L378) as well as the uncertainty propagation method (L630 – L646).

Comment 9

4) It was not possible to completely follow the responses to previous review that was available, as no prior version was supplied, and would be beyond the scope of such a review. However, I agree with referee 1 that the manuscript even in this revised version is rather opaque. Hopefully the comments above help in a thorough re-writing that clarify the arguments made.

We hope that the implemented changes in: 1.) interpretation of the results in context of clustering (comment 4 and 5), and 2.) updated definitions and consistent terminology on MAOC capacity (comment 6) have improved the readability of the manuscript. In line with this comment, we have also revised some sections of the manuscript to facilitate interpretation. For example, we discussed the sensitivity analyses (L213 – L217) and uncertainty propagation presented in this response letter (L376-L378, L630 – L646) as well as aligning the names of the regression methods with the previous literature. What was previously referred to as Hassink's method (Hsat) is actually an altered methodology of Hassink. The method estimates the boundary line (BL) by linear quantile regression (Feng et al. 2013). We have corrected this and renamed the other alternative methods as variations: piece-wise boundary line (PBL) and non-

linear boundary line (NBL). We have also added additional text within the abstract to underline the main findings of the SOC risk index (L25-L27).

Comment 10

Reviewer #3 (Remarks on code availability):

n/a

We have now made the code available under this link, and can be reviewed. The repository is restricted for the duration of the review process.

Comment 11

Reviewer #4 (Remarks to the Author):

We thank the reviewer for their comments to improve our manuscript.

Reviewer #4 (Remarks to the Author):

Thank you for the revisions.

I focus on these comments on the changes made in response to my earlier comments, and any new writing.

Comment 1

I note that while the revisions are attempting to address the concerns and the responses are addressing much of the suggestions, the reader does not fully benefit as they are not fully implemented in the manuscript (a typical response is (not verbatim): yes, this is a challenge, but here is a reason why it is not a big problem and now we ignore it). I also note that inaccurate language creeps in, which would be possible to address and should ideally not be present at this stage of review&revision,- which makes is a bit challenging looking forward to an eventually near-flawless manuscript. I give the authors benefit of the doubt and hope for their own sake that they do not take the revisions lightly (which I think they did not do in the rebuttal, but seemed to not have taken to heart in the revised manuscript).

We thank the reviewer for the comments, and we want to reassure that we took the previous comments into high consideration. In fact, the error propagation required a lot of effort to calculate the uncertainty of our estimates based on hundreds of model runs. Methods are fully reported in the supplementary and part in the main text. We agree that we did not explicitly report the interval of confidence (5th and 95th quantiles, reported in the Supplementary Table 3) for the areas of each SOC risk class (see also Comment #5). Therefore, we have decided to move what was Supplementary Table 3 in the main text (now Table 1) and we reference it when discussing the associated uncertainty with each risk index class (L383-392).

In our previous response, we also performed the sensitivity analysis for assessing the influence of cluster definitions as requested (L212-218, in the main text). Again, the results are described in the supplementary material for preserving the flow in text, such that we discuss the main ideas and outcomes. Here we also took into consideration the constraints on the length of the manuscript.

We have also made the code of the original analysis available as requested, including the two major changes based on the previous review (uncertainty propagation and sensitivity analysis). We are grateful for the recommendations that we found very useful to further investigate the robustness of our work.

Detailed Comments

Comment 2

Line 202: what is a “organo-organo C bonds on the clay surface”? Either it is an organo-organic C bond or it is an organo-mineral bond, but a “organo-organo C bonds on the clay surface” seems a

contradiction,- or they mean a organo-organic bond or OC on a clay surface. Please scrutinize your phrasing and make sure it is not confusing or inaccurate. (also L 248)

We thank the reviewer for pointing out this inconsistency. We have corrected this, see L80-81 for a better descript and L249 for a correction.

Comment 3

Line 203: “particularly for data low in fine fraction content” does also not make sense without some modification. How can data be low in fine fraction? Do you mean a dataset of soils are low in fine fractions? Or something else? Unfortunately a few of these inaccurate remarks are creeping in and I hope you or your co-authors can rectify it.

We appreciate the detail in which the reviewer has assessed the text and corrected this. Together with our revisions to improve readability (comment #9), we have also revised these kinds of linguistic inconsistencies throughout the text.

Comment 4

Line 246: “Differences in the PBL method” should probably read “Differences between the PBL compared to the BL method”.

Corrected.

Comment 5

Line 279: “We anticipate that these limitations might affect the estimate of the maximum MAOC capacity for fine-textured soils in some pedo-climatic clusters. Nonetheless, the mean”: I appreciate that the authors followed up with some error calculations, but then they do not seem carry that over to the manuscript, and do not give quantitative information about how reliable the estimates are, but rather argue “...align very closely with...” without giving the reader the benefit of their estimates, but leave this referee ignored in outlining the error range for in this case the maximum MAOC; there is some guidance in the replies of the authors that would be helpful to also see in the published manuscript. If that could be done in lucid language this would be wonderful.

We thank the reviewer for this comment, and we are very sorry they felt ignored, it was by no means our intention. In fact, our original intent, to address the previous reviewer’s concern, was to describe the anticipated effect of excluding soil under oxygen limited conditions, geogenic carbon and other soils underrepresented in LUCAS. Based on review round 2, we did a comparison with legacy data (Suppl. Figs. 5-9) and noticed this effect. We have now clarified the description (L279-L281).

Regarding the uncertainty propagation. In our opinion, it is more valuable when the error associated with MAOC predictions from VNIR have been propagated for the entire analysis (calculation of effective MAOC capacity, calculation of MAOC saturation by fine fraction bin and

the SOC risk index). Since the variation in MAOC would affect all these steps and is thus captured in the variation of the SOC risk classes.

We see now, and agree with the reviewer, that all the work we did for estimating the uncertainty was not capitalized in the main text. To solve this, we have now moved Table 3 from the supplementary material to the main text (now Table 1) that details the effect of the uncertainty propagation analysis (L383-L392). This now gives an overview of the variation in the area (Mha) of the different risk index classes, given the associated uncertainty with MAOC.

Comment 6

Line 300: Rephrase to “soils low in fine fraction content.” (data cannot be low in fine fraction)

Corrected.

Comment 7

Line 302: How many did exceed 100% in BL and NBL? Give numbers in this sentence.

Added, see L302 and L305.

Comment 8

Line 315: I guess that you introduce the term and concept “effective MAOC capacity” here? At least, I was not able to find the term earlier in the manuscript. What is that? (referring to the methods is not sufficient) How is it calculated? It is really good if you can write the text in a way that it can be understood without extensive supplementary reading. For example, if this is a new term in the literature or you define it in a new way, then you would want to choose words such as “we establish”; if it has not been used before in this manuscript, then at least you may want to say “we calculated”; what you write makes the reader think that this is a well known term or has been mentioned before in the text, or both,- which does not seem to be the case. (the same applies to Δ SOC in Line 349, which was also not mentioned before) This may require some significant restructuring.

Indeed, the definition was in line 85-86 where from Line 64 onwards we referenced the literature to introduce the concept (e.g. Ingram and Fernandes). The introduction in L66-L84 aims to explain the transition from the maximum theoretical to the effective MAOC capacity. We have now modified the text to clarify this and contextualize our clustered approach to calculate the effective MAOC capacity (L85-93). The calculation of the effective MAOC capacity is also now better described before the results are introduced (L197-206) and in more detail in the methods section (L577-601, unchanged in this revision).

Comment 9

Line 320: I appreciated the strong methodological pitch in this paragraph, but it did not contribute to readability,- rather it made this reader wonder whether there is anything to learn about SOC

beyond differences in computational approaches. This seems to be somewhat an issue throughout: not very clear in terms of storytelling that is buried in methodological jargon. That can be fixed, and should be fixed for an interdisciplinary journal with a broad audience.

We have now inverted this sentence to increase readability, we have also added additional changes for readability or elaborated the text to facilitate understanding by the reader. Changes to readability were mainly added to the introduction and additional context was added on L100-104, L201-206, L216-218, L312-L323, 341-346.

Comment 10

Line 332: Capitalize “findings”

Done.

Comment 11

The term “SOC Risk index” (Line 309) would benefit from a more intuitive description in lines 310-319 to facilitate understanding.

We agree that the transition was rather abrupt to the SOC risk index. We have now changed the text such that the SOC risk index is introduced in a more comprehensive manner, see L312-L323. We have also provided a more elaborate description of the risk index in the introduction, see L100-104.

Comment 12

I hope you find these suggestions useful to make your paper accessible, and look forward to seeing it in print.

We thank the reviewer for the detailed comments to improve our manuscript.

Reviewer #5 (Remarks to the Author):

Comment 13

We thank the reviewer for the detailed comments to improve our manuscript.